



# Contrasting dynamics of hydrological processes in the Volta River basin under global warming

Moctar Dembélé[1,2,7], Mathieu Vrac[3], Natalie Ceperley[7], Sander J. Zwart[1], Josh Larsen[4], Simon J. Dadson[5,6], Grégoire Mariéthoz[2] & Bettina Schaefli[2,7]

[1]1International Water Management Institute, CSIR Campus, No. 6 Agostino Neto Road, Accra, Ghana
[2]Institute of Earth Surface Dynamics (IDYST), Faculty of Geosciences and Environment, University of Lausanne, 1015 Lausanne, Switzerland
[3]Laboratoire des Sciences du Climat et de l'Environnement (LSCE-IPSL), CEA/CNRS/UVSQ, Université Paris-Saclay Centre d'Etudes de Saclay, Orme des Merisiers, 91191 Gif-sur-Yvette, France
[4]School of Geography, Earth and Environmental Sciences, University of Birmingham, Birmingham, UK
[5]School of Geography and the Environment, University of Oxford, South Parks Road, Oxford OX1 3QY, UK
[6]UK Centre for Ecology and Hydrology, Wallingford, Oxfordshire OX10 8BB, UK
[7]Institute of Geography (GIUB) and Oeschger Centre for Climate Change Research (OCCR), University of Bern, 3012 Bern, Switzerland

*Correspondence to*: Moctar Dembélé (moctar.dembele@cgiar.org)

**Abstract.** A comprehensive evaluation of the impacts of climate change on water resources of the West Africa Volta River basin is conducted in this study, as the region is expected to be hardest hit by global warming. A large ensemble of twelve general circulation models (GCM) from CMIP5 that are dynamically downscaled by five regional climate models (RCM) from CORDEX-Africa is used. In total, 43 RCM-GCM combinations are considered under three representative concentration pathways (RCP2.6, RCP4.5 and RCP8.5). The reliability of each of the climate datasets is first evaluated with satellite and reanalysis reference datasets. Subsequently, the Rank Resampling for Distributions and Dependences (R2D2) multivariate bias correction method is applied to the climate datasets. The corrected simulations are then used as input to the fully distributed mesoscale Hydrologic Model (mHM) for hydrological projections over the twenty-first century (1991-2100).

Results reveal contrasting changes in the seasonality of rainfall depending on the selected greenhouse gas emission scenarios and the future projection periods. Although air temperature and potential evaporation increase under all RCPs, an increase in the magnitude of all hydrological variables (actual evaporation, total runoff, groundwater recharge, soil moisture and terrestrial water storage) is only projected under RCP8.5. High and low flow analysis suggests an increased flood risk under RCP8.5, particularly in the Black Volta, while hydrological droughts would be recurrent under RCP2.6 and RCP4.5, particularly in the White Volta. Disparities are observed in the spatial patterns of hydroclimatic variables across climatic zones, with higher warming in the Sahelian zone. Therefore, climate change would have severe implications for future water availability with concerns for rain-fed agriculture, thereby weakening the water-energy-food security nexus and amplifying the vulnerability of the local population. The variability between climate models highlights uncertainties in the projections and indicates a need to better represent complex climate features in regional models. These findings could serve as a guideline for both the scientific





community to improve climate change projections and for decision makers to elaborate adaptation and mitigation strategies to
cope with the consequences of climate change and strengthen regional socio-economic development.

## 1 Introduction

Climate warming is projected to occur at a faster rate in West Africa than the global average during the twenty-first century
(Todzo et al., 2020). Anthropogenic greenhouse gas emissions have led to an unprecedented increase in surface air temperature,
which has resulted in the intensification of the hydrological cycle (Sylla et al., 2016). Therefore, recurrent floods and droughts
could persist in the future because rainfall is projected to decrease in frequency but increase in intensity (Aich et al., 2014;Dosio
et al., 2020). In the face of climate change and variability, West Africa is particularly vulnerable because of its high reliance
on rain-fed agriculture and limited institutional capacities to cope with climate change and variability (Karambiri et al.,
2011;Sultan and Gaetani, 2016;Yira et al., 2017). Climate change and anthropogenic pressures pose an increasing stress on
water resources (Sood et al., 2013). Freshwater shortages that lead to a decline in basin-scale irrigation water availability could
have dire consequences for sustainable agriculture (Sylla et al., 2018b). Consequently, global warming is a serious threat for
water and food security in West Africa. However, addressing this threat is currently difficult given the high variability in
climate projections, which can impose very different hydrological implications for West Africa (Dosio et al., 2020), thus
imposing greater urgency for further investigations on the impacts of climate change on hydrological processes.

In the transboundary Volta River Basin (VRB) located in West Africa, water resources are fundamental for agriculture,
hydropower generation, fisheries and other ecosystem services (Williams et al., 2016). Most of the agriculture is rain-fed but
many regions rely on irrigated agriculture (Roudier et al., 2014). Hydropower is a major source of electricity production with
the potential to grant more access to energy in the region (Kling et al., 2016;Stanzel et al., 2018). Future water resource
developments in the VRB focus primarily on hydroelectricity and irrigation (McCartney et al., 2012). However, severe impacts
of climate change on water resources in the VRB will impede future socio-economic development (Sood et al., 2013).
Anticipation of the future evolution of the hydrological cycle in the VRB is essential to ensuring the adaptive capacities of the
riparian countries to the regional consequences of global warming (Jin et al., 2018). However, there is relatively little
knowledge of the impacts of climate change on the future water resources in West Africa in general (e.g., Kasei, 2010;Oyerinde
et al., 2016;Yira et al., 2017), and only a few studies targeted the VRB (Jung et al., 2012;Okafor et al., 2019;Roudier et al.,
2014). These studies usually focused on climatic variables (i.e. precipitation and temperature), and when considering
hydrological modelling, they focused usually on one variable (e.g. streamflow). The limitations of climate change impacts
studies on water resources in West Africa arise from the lack of hydrological and meteorological observations to drive models,
in addition to uncertainties related to climate projection data as well as hydrological models (Dembélé et al., 2019;Oyerinde
et al., 2016;Sidibe et al., 2020;Sylla et al., 2018a). Despite considerable progress in improving climate projections and the
efforts in investigating climate change in West Africa (e.g., Diallo et al., 2016;Kebe et al., 2017;Mahé et al., 2013;Nikiema et



al., 2017), the need for understanding the future evolution of the hydrological cycle under varying scenarios still exists (Eyring et al., 2019;Sidibe et al., 2020;Sylla et al., 2016).

The goal of this study is to analyse repercussions of projected changes of rainfall and temperature on the seasonal and annual trends of various components of the hydrological cycle (i.e., streamflow, total runoff, potential and actual evaporation, groundwater recharge, soil moisture and terrestrial water storage) and water availability in the twenty-first century, under

multiple future scenarios. Therefore, this study investigates the dynamics of various hydrological processes in the VRB under a changing climate and the implications for water availability and extreme events like floods and droughts. The results can provide knowledge to inform research and water planning to develop and implement adaptation and mitigation measures that alleviate the impacts of global warming in the VRB.

## 2 Study area

The Volta River Basin (VRB) is a major transboundary basin in West Africa (Figure 1), and the ninth largest drainage basin in sub-Saharan Africa (UNEP-GEF, 2013). It is located between latitudes 5°40' N and 14°55' N and longitudes 5°25' W and 2°20' E, and covers approximately 415,600 km$^2$ shared among six countries (Benin, Burkina Faso, Côte d'Ivoire, Ghana, Mali, and Togo). The population of the VRB was estimated to 23.8 million people in 2010 and it is projected to reach 56.1 million in 2050 (Williams et al., 2016). The altitude ranges from zero to 940 m a.s.l., but the topography is predominantly flat with a

mean altitude of about 255 m a.s.l., as most of the basin lies below 400 m a.s.l. (Dembélé et al., 2020c). The land cover of the basin area is composed of savannah (75%), cropland (13%), forest (9%), water bodies (2%) and bare land and settlements (1%). The climate is driven by the latitudinal and seasonal oscillation of the Inter-Tropical Convergence Zone (ITCZ), which governs rainfall occurrence in the VRB. Rainfall depicts a south-north gradient of increasing aridity, with four eco-climatic zones (i.e. Sahelian, Sudano-Sahelian, Sudanian and Guinean), and it is characterized by interannual and multidecadal

variabilities (Nicholson et al., 2018). The Volta River flows north-south over 1,850 km and drains into the Atlantic Ocean at the Gulf of Guinea after transiting into the Lake Volta formed by the Akosombo dam. The drainage system is composed of four sub-basins known as Black Volta (152,800 km$^2$), White Volta (113,400 km$^2$), Oti (74,500 km$^2$) and Lower Volta (74,900 km$^2$) (Dembélé, 2020).

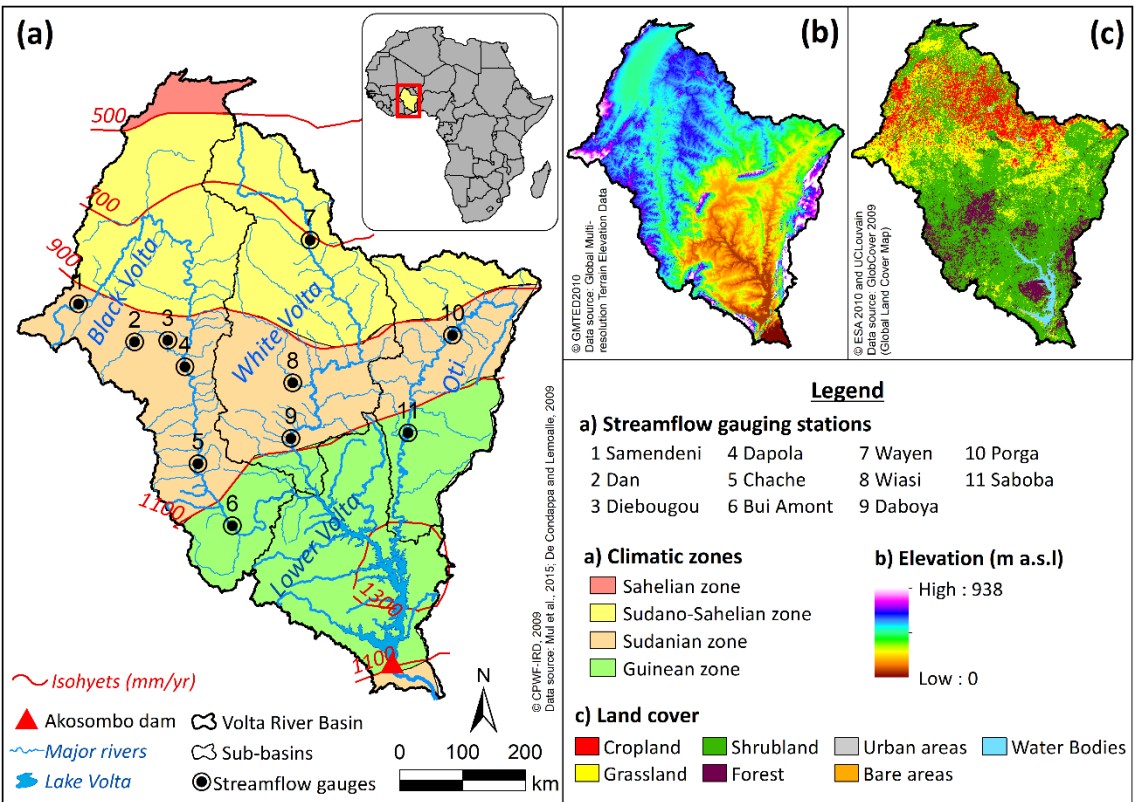

**Figure 1. Physical and hydroclimatic characteristics of the Volta River basin. (a) hydrographic network and climatic zones, (b) digital elevation model, (c) land cover.**

## 3 Data and Methods

### 3.1 Overview of the methodology

The methodology adopted for the assessment of climate change impacts on water resources in the VRB is summarized in Figure 2. The main steps consist of the bias correction of the RCMs forced by the GCMs, the modelling of hydrological processes based on the climate projection datasets, and the analysis of the future changes in the modelled hydrological processes.

Uncertainties in the climate projections are addressed by employing a large ensemble of twelve General Circulation Models (GCMs) downscaled by five Regional Climate Models (RCMs) under three Representative Concentration Pathways (RCPs; Moss et al., 2010;Van Vuuren et al., 2011). The RCMs are obtained from the Coordinated Regional-climate Downscaling Experiment (CORDEX) for Africa (Giorgi et al., 2009). Only considering the highest RCP8.5 scenario as the "business as usual" scenario in climate change studies is increasingly criticized because the assumption of the heavy use of coal in RCP8.5 is unrealistic (Hausfather and Peters, 2020;Ritchie and Dowlatabadi, 2017). However, the current emissions are found to be in





line with the RCP8.5 scenario (Peters et al., 2013), and there are suggestions for giving RCP8.5 a high priority (O'Neill et al.,
2016).


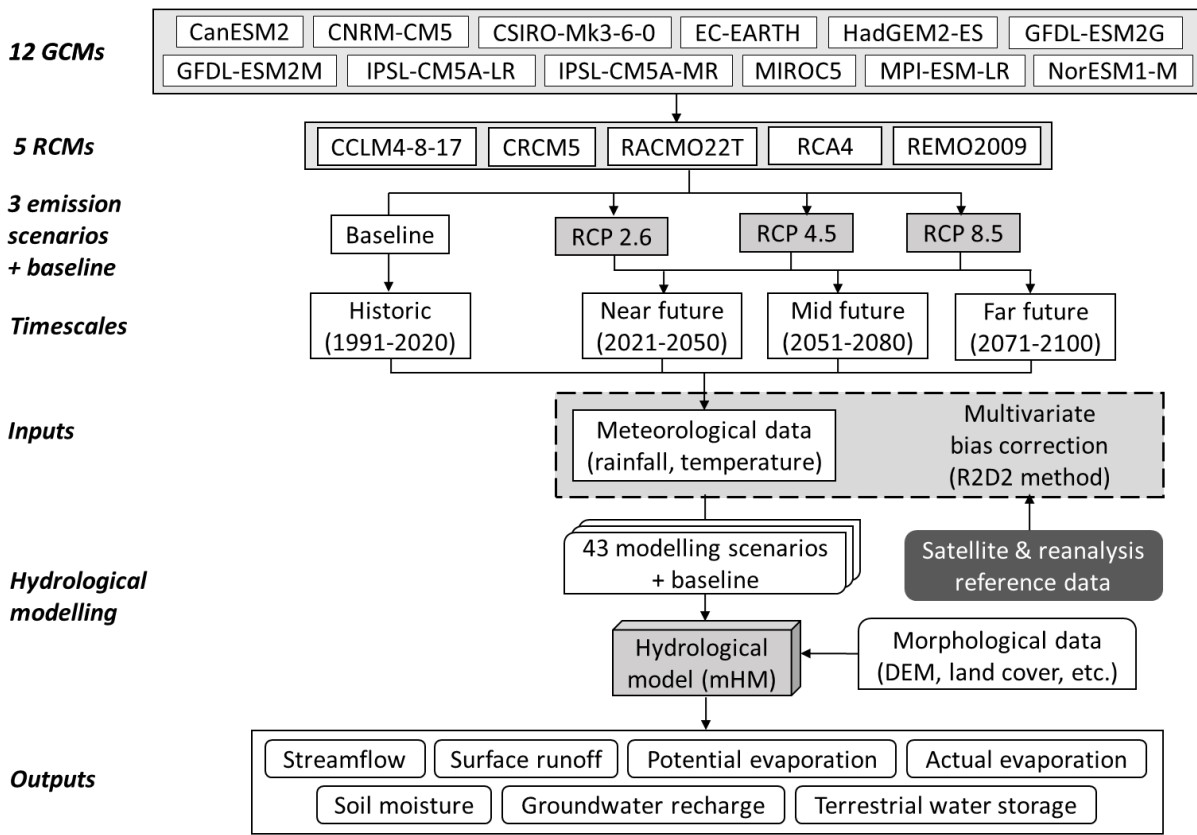

**Figure 2. Overview of the procedure for assessing the impacts of climate change on hydrological processes.**

The RCM-GCM datasets are first evaluated by comparing them to the best performing satellite and reanalysis datasets of
rainfall and temperature for hydrological modelling in the VRB. Subsequently, a multivariate bias correction is applied to the
climate projection datasets using the Rank Resampling for Distributions and Dependences (R2D2) method (Vrac and Thao,
2020). Finally, the bias corrected climate projection datasets are used as input in the fully distributed mesoscale Hydrologic
Model (mHM) to assess the impact of climate change on multiple hydrological fluxes and state variables. Although the
performance of a model during past and present conditions does not guarantee its reliability for future projections (Stanzel et
al., 2018), having a well performing model that simulates realistic hydrological processes is a prerequisite for any sound impact
study (Krysanova et al., 2018).



## 3.2 Climate projection datasets

CORDEX generates high-resolution historical and future climate projections for regional domains, by downscaling GCMs
participating in the fifth Coupled Model Intercomparison Project (CMIP5; Taylor et al., 2012). Based on data availability on
the Earth System Grid Federation (ESGF) platform (https://esgf-data.dkrz.de, last accessed on 22.03.2020), twelve GCMs
from CMIP5 dynamically downscaled with five RCMs available from the CORDEX-Africa initiative are selected for this
study (Table 1). The spatial resolution of the RCMs is 0.44° (∼ 50 km) in latitude and longitude. As not all GCMs are
downscaled by the RCMs, there are 21 RCM-GCM combinations available for the global runs in the historical period (1951-
2005), while 43 RCM-GCM combinations are available for the future projections (2006-2100) under various representative
concentration pathways (RCP2.6, RCP4.5 and RCP8.5). The RCPs correspond to different scenarios of future concentrations
and emissions of greenhouse gases and air pollutants, and land-use change until 2100, relative to the pre-industrial times (Moss
et al., 2010;Van Vuuren et al., 2011). The three RCPs used in this study are (i) RCP2.6 corresponding to a mitigation scenario
with a very low radiative forcing level that peaks at ~3 W m$^{-2}$ (~490 ppm $CO_2$ equivalent) and declines to ~2.6 W m$^{-2}$ by 2100;
(ii) RCP4.5 representing a medium stabilization scenario without overshoot pathway to 4.5 W m$^{-2}$ (~650 ppm $CO_2$ equivalent)
at stabilization after 2100; and (iii) RCP8.5 corresponding to a very high emission scenario with rising radiative forcing
pathway leading to 8.5 W m$^{-2}$ (~1370 ppm $CO_2$ equivalent) by 2100. Approximatively, RCP2.6 corresponds to a +2 °C global
temperature stabilization by 2100, while RCP4.5 and RCP8.5 correspond to +3.5 °C and +5 °C increase in global temperature
by 2100 (Konapala et al., 2020). The climate variables used in this study are daily data of rainfall and minimum, maximum
and average air temperature, which are obtained for the climate model realization r1i1p1.

**Table 1. List of the CORDEX-Africa General Circulation Models (GCMs) datasets downscaled by Regional Climate Models (RCMs) and their availability per Representative Concentration Pathways (RCP) marked with a cross (x).**

| RCMs | GCMs | RCP2.6 | RCP4.5 | RCP8.5 |
|------|------|--------|--------|--------|
| CCLM4-8-17 | CNRM-CM5 | | x | x |
| | HadGEM2-ES | | x | x |
| | MPI-ESM-LR | | x | x |
| CRCM5 | CanESM2 | | x | |
| | MPI-ESM-LR | | x | |
| RACMO22T | EC-EARTH | | x | x |
| RCA4 | CanESM2 | | x | x |
| | CNRM-CM5 | | x | x |
| | CSIRO-Mk3-6-0 | | x | x |
| | EC-EARTH | | | x |
| | IPSL-CM5A-MR | | x | x |
| | MIROC5 | x | x | x |
| | HadGEM2-ES | x | x | x |
| | MPI-ESM-LR | x | x | x |
| | NorESM1-M | x | x | x |



| | | | | |
|---|---|---|---|---|
| | GFDL-ESM2M | | x | x |
| | IPSL-CM5A-LR | x | | x |
| | MIROC5 | x | | x |
| REMO2009 | HadGEM2-ES | x | | x |
| | MPI-ESM-LR | x | x | x |
| | GFDL-ESM2G | x | | |

### 3.3 Multivariate bias correction of climate data

Assessing the reliability of climate projections data in reproducing observations is a precondition to impact studies (Eyring et al., 2019). However, observations are also subject to uncertainties. To address these uncertainties, the climate projection datasets are evaluated with respect to ten satellite-based and reanalysis-based rainfall datasets and six temperature datasets (Table 2). The selected satellite-based and reanalysis-based rainfall datasets demonstrated the best performances for large-scale hydrological modelling in the VRB as shown by Dembélé et al. (2020c). Hereafter, the datasets composed of both satellite and reanalysis products will be referred to as *observations*.

Table 2. Satellite-based and reanalysis-based rainfall and temperature products for climate projection data evaluation. *P*: rainfall, *T*: temperature, NP: near-present.

| Datasets | Name/ website | Used variables | Temporal coverage | References |
|---|---|---|---|---|
| **TAMSAT v3.0** | Tropical Applications of Meteorology using SATellite (TAMSAT), African Rainfall Climatology and Time-series (TARCAT) https://www.tamsat.org.uk/data/archive | *P* | 1983-NP | Maidment et al. (2017), Tarnavsky et al. (2014) |
| **CHIRPS v2.0** | Climate Hazards group InfraRed Precipitation with Stations (CHIRPS) V2.0 http://chg.ucsb.edu/data/chirps/ | *P* | 1981-NP | Funk et al. (2015) |
| **ARC v2.0** | Africa Rainfall Estimate Climatology (ARC 2.0) https://www.cpc.ncep.noaa.gov/products/international/data.shtml | *P* | 1983-NP | Novella and Thiaw (2013) |
| **MSWEP v2.2** | Multi-Source Weighted-Ensemble Precipitation (MSWEP) V2.2 http://www.gloh2o.org/ | *P* | 1979-NP | Beck et al. (2017) |
| **PERSIANN-CDR v1r1** | Precipitation Estimation from Remotely Sensed Information using Artificial Neural Networks (PERSIANN) Climate Data Record (CDR) V1R1 http://chrsdata.eng.uci.edu/ | *P* | 1983-2016 | Ashouri et al. (2015) |
| **WFDEI-CRU** | WATCH Forcing Data ERA-Interim (WFDEI) corrected using Climatic Research Unit (CRU) dataset www.eu-watch.org | *P* | 1979-2018 | Weedon et al. (2014) |
| **WFDEI-GPCC** | WATCH Forcing Data ERA-Interim (WFDEI) corrected using Global Precipitation Climatology Centre (GPCC) dataset ftp://rfdata:forceDATA@ftp.iiasa.ac.at/ | *P, T* | 1979-2016 | Weedon et al. (2014) |
| **PGF v3** | Princeton University global meteorological forcing (PGF) http://hydrology.princeton.edu/data/pgf/ | *P, T* | 1948-2012 | Sheffield et al. (2006) |
| **ERA5** | European Centre for Medium-range Weather Forecasts ReAnalysis 5 (ERA5) hourly data on single levels https://cds.climate.copernicus.eu/ | *P, T* | 1979-NP | Hersbach et al. (2020) |
| **MERRA-2** | Modern-Era Retrospective Analysis for Research and Applications 2 (rainfall: M2T1NXFLX_V5.12.4; temperature: M2SDNXSLV_V5.12.4) https://disc.gsfc.nasa.gov/datasets/ | *P, T* | 1980-NP | Gelaro et al. (2017), |



| | | | | Reichle et al. (2017) |
|---|---|---|---|---|
| **EWEMBI v1.1** | EartH2Observe, WFDEI and ERA-Interim data Merged and Bias-corrected for ISIMIP (EWEMBI) http://doi.org/10.5880/pik.2016.004 | *T* | 1976-2013 | Lange (2016) |
| **JRA-55** | Japanese 55 year ReAnalysis (JRA-55); rainfall: fcst_phy2m125; temperature: anl_surf125 https://jra.kishou.go.jp/JRA-55/index_en.html | *T* | 1959-NP | Kobayashi et al. (2015) |

As discrepancies are observed between the cumulative distribution functions of the observations and the climate projection datasets (Figure 3), a bias correction is applied before using the climate datasets for hydrological modelling, as usually recommended (Hakala et al., 2018;Teutschbein and Seibert, 2012). The R2D2 method (Vrac and Thao, 2020) is adopted for a multivariate bias correction of the climatic variables. R2D2 is a rank analogue-based method that adjusts not only the univariate distributions of climatic variables, but also the inter-variable and inter-site dependence structures (Vrac, 2018). Bias correction with the R2D2 approach is achieved in two steps. First, the marginal distributions of univariate time series are adjusted using any univariate bias correction method. Here, the "cumulative distribution function transform" (CDF-t) approach (e.g., Vrac et al., 2012) is used to adjust the marginal properties of the univariate time series. Second, R2D2 is used to adjust the dependence structure between several variables, independently of their marginal distribution (i.e. empirical copula function). R2D2 has performed better compared to other bias correction methods as demonstrated by François et al. (2020).

In the current multivariate bias correction, rainfall and temperature datasets are corrected simultaneously to preserve the temporal and spatial dependences between the climatic variables. The bias correction is done using only one of the observational datasets as reference data, the WFDEI, because it has both rainfall and temperature data over a long period (1979-2016). Furthermore, the WFDEI dataset previously demonstrated good performances in the VRB (Dembélé et al., 2020b). Therefore, it is chosen as reference data to limit uncertainties in the bias correction (Tarek et al., 2021). The period 1981-2005 is taken as the reference period for training the bias correction of the climate projection datasets, whose time series are divided into several 25-year periods over the period 1951-2100 to correspond to the length of the reference period. The multivariate bias correction is applied by grouping the data per calendar month in each sub-period of 25 years, which improves seasonality in the corrected data.

### 3.4 Hydrological Modelling

### 3.4.1 Model set-up and modelling period

The fully distributed mesoscale Hydrologic Model (mHM) simulates dominant hydrological processes with seamless spatiotemporal patterns in the modelling domain (Kumar et al., 2013;Samaniego et al., 2010;Samaniego et al., 2017). Telteu et al. (2021) provide a description of the formulation of hydrological processes in mHM. The model configuration is similar to the study of Dembélé et al. (2020b), which provides full details on the model setup, calibration, evaluation and performance across scales. The study also demonstrates the ability of the mHM model to reproduce reliable spatiotemporal patterns of





various hydrological processes after a robust multivariate model calibration with streamflow and satellite data of evaporation,

soil moisture and water storage.

The calibrated model is run over the entire data availability period (1951-2100) of the RCM-GCM datasets, with 1951-1960 as spin-up period. The baseline or historical period for climate change impact assessment is 1991-2020. Projections are assessed for the near-term future (2021-2050), the long-term future (2051-2080) and the late-century (2071-2100). In total, 21 RCM-GCM combinations are available for the historical runs, while for future projections, 9 RCM-GCM combinations are

available for the RCP2.6, 16 for RCP4.5 and 18 for RCP8.5 (Table 1). Although future land use and land cover (LULC) scenarios are not used in this study, the temporal dynamic of LULC is accounted for by using different maps over the modelling period. Based on high-resolution LULC data available between 1992 and 2015 from the European Space Agency Climate Change Initiative (ESA, 2017), LULC maps for 1992, 2005 and 2015 are used for the periods 1951-1990, 1991-2020 and 2021-2100, respectively.

**3.4.2 Model reliability**

The realism of the hydrological simulations is verified with the Budyko framework (Budyko, 1974), which helps to estimate mean annual water availability as a function of aridity. The supply-demand framework, which is increasingly used in hydrological modelling (Greve et al., 2020;Wang et al., 2016), is valid for large catchments under steady state, considering long-term water balance and energy balance (Donohue et al., 2010;McVicar et al., 2012). The exercise consists of verifying if

the ratio of the long-term mean annual potential evaporation to precipitation (aridity index) and the ratio of long-term mean actual evaporation to precipitation (evaporative index) of the climate projections are coherent with the energy and water limits for a given climate (Sposito, 2017;Donohue et al., 2011). Here, the term evaporation involves all sources of evaporation including transpiration (Miralles et al., 2020;Savenije, 2004). The Budyko curve is formulated with equation 1, after Budyko (1974), as follows:

$$\frac{\overline{E_a}}{\overline{P}} = \left[ \phi \tanh\left(\frac{1}{\phi}\right) \left(1 - \exp(-\phi)\right) \right]^{1/2} \tag{1}$$

where $\overline{P}$ is the long-term mean annual precipitation, $\overline{E_a}$ is the long-term mean annual actual evaporation and $\phi$ is the aridity index.

Uncertainties in the model inputs and outputs are assessed in terms of variability between the simulations corresponding to different climate models using the second order coefficient of variation ($V_2$), which addresses the limitations of the classical coefficient of variation (Kvålseth, 2017), and is defined as follows:

$$V_2 = \left( \frac{s^2}{s^2 + \bar{x}^2} \right)^{1/2} \tag{2}$$

where $s$ is the standard deviation and $\bar{x}$ is the mean of a sample data $x = (x_1,\dots, x_n) \in R^n$. $V_2$ represents the distance between $x$ and $\bar{x}$ relative to the distance between $x$ and the origin zero, and it varies from 0 to 1 or 0% to 100%.





Finally, the percentage of RCM-GCM agreement is calculated as the number of models agreeing on the same direction of change (i.e. increase or decrease) relative to the total number of models, which shows the robustness of the ensemble of climate projections.

**4 Results**

**4.1 Multivariate bias correction**

The raw RCM-GCM datasets are evaluated by comparing their cumulative distribution functions to those of the observations over the period 1983-2005 corresponding to the concomitant availability period of all the observation datasets (Figure 3). The distribution of most of the raw RCM-GCM datasets presents discrepancies with the observations, with larger gaps for 210 temperature than rainfall. The multivariate bias correction with the R2D2 method visually performs well by adjusting the distributions of the RCM-GCM datasets to the WFDEI reference dataset for all the climatic variables. Therefore, the corrected RCM-GCM datasets are expected to provide reliable hydrological simulations in the VRB.





**Figure 3. Cumulative distribution functions (CDF) of daily rainfall (*P*) and average, maximum and minimum daily air temperature**
**(*T*avg, *T*max and *T*min) before and after multivariate bias correction of various RCM-GCM datasets, over the 1983-2005 period. The black line and grey-shaded area represent the mean and the 90% confidence interval of the satellite and reanalysis datasets of rainfall (10 datasets) and temperature (6 datasets).**

## 4.2 Plausibility of hydrological simulations

The general plausibility, here used to mean water and energy balance consistency within the Budyko framework, of the
hydrological simulations using various RCM-GCM datasets as inputs to the mHM model under various RCPs and various periods is illustrated in Figure 4.





All the RCM-GCM datasets provide plausible hydrological simulations, at least in terms of water and energy balance, as they respect the water and energy limits imposed within the Budyko framework. On average, the evaporative index ($E_a/P$) is between 0.86 and 0.97, while the aridity index ($E_p/P$) is between 2.2 and 4.4, which corresponds to expected values for sub-humid and semi-arid environments such as the VRB (Gunkel and Lange, 2017). It is noteworthy that future projections are shifted towards a lower evaporative index and they large model dependent variability in aridity ranges, particularly under RCP4.5 and RCP8.5.

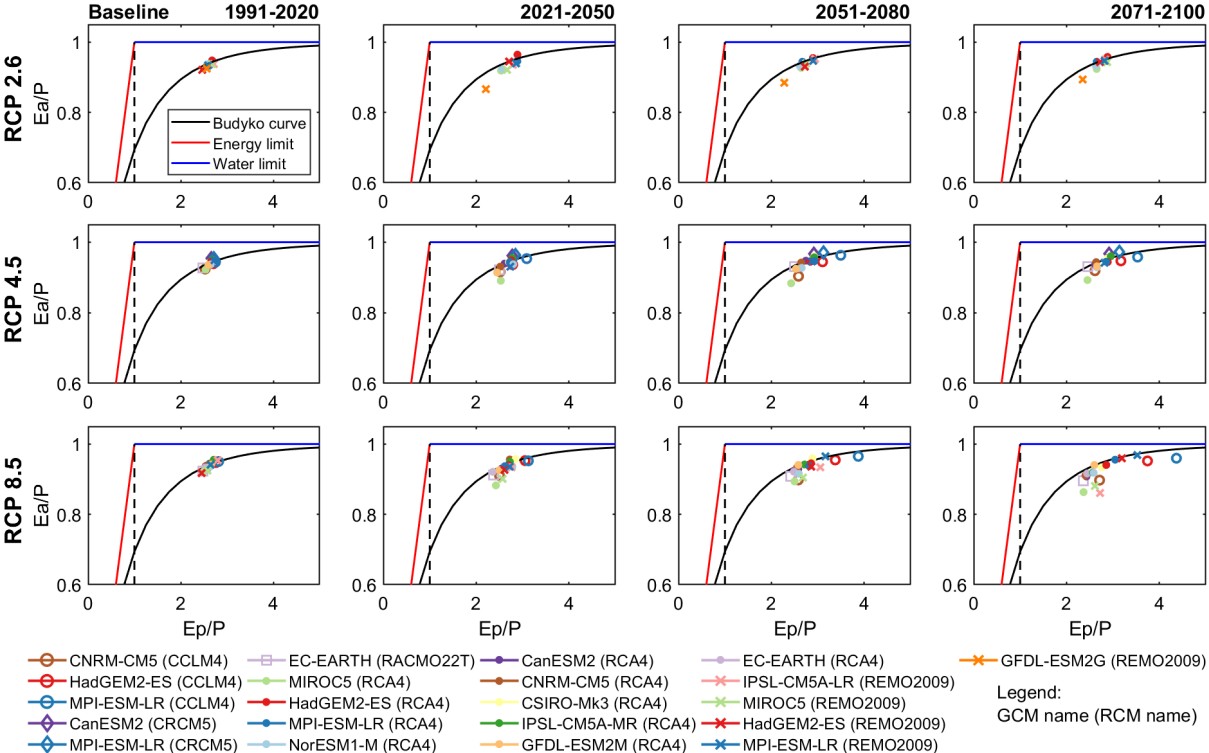

**Figure 4. Plausibility of hydrological processes with the Budyko framework representing the evaporative index, i.e. ratio of actual evaporation to rainfall ($E_a/P$), as a function of the aridity index, i.e. ratio of potential evaporation to rainfall ($E_p/P$). The black dashed line represents the limit where precipitation equals potential evaporation.**

### 4.3 Seasonal changes in hydroclimatic variables

The annual cycles of climatic and hydrological variables are illustrated in Figure 5 for RCP 8.5 (see Figures S1-S10 for other RCPs). The hydroclimatic variables analysed are rainfall ($P$), average air temperature ($T_{avg}$), potential evaporation ($E_p$), actual evaporation ($E_a$), root-zone soil moisture ($S_u$), terrestrial water storage ($S_t$), total runoff ($Q_{run}$) and groundwater recharge ($R_r$). It is noteworthy that $T_{avg}$ and $E_p$ show a clear bimodal annual cycle in the VRB, with the first mode peaking between March and April, corresponding to the hottest period, while the second mode starts in September and peaks around October and November. For the hydroclimatic variables depicting a unimodal annual cycle, the peak of the cycle is observed around August and September, corresponding to the rainiest period.





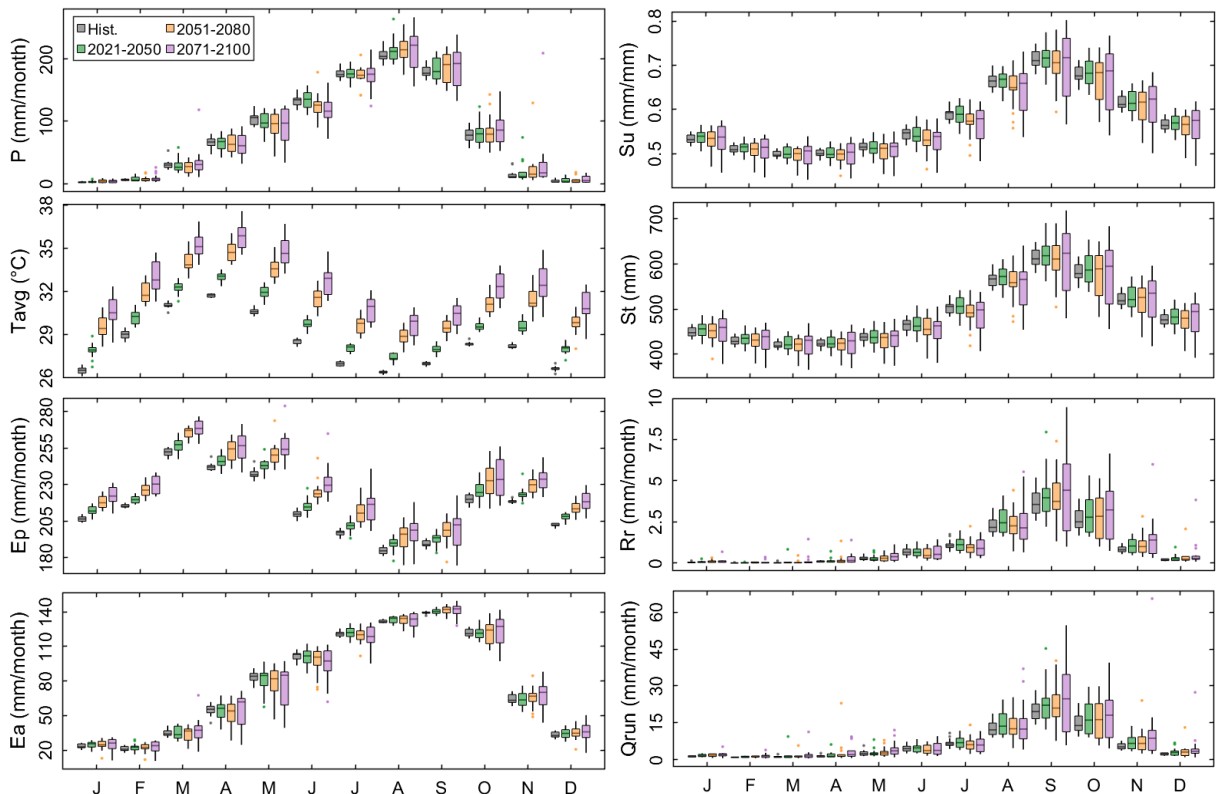

**Figure 5. Annual cycles of climatic and hydrological variables under RCP8.5 over the historical and future periods. Each boxplot represents the spread among the RCM-GCMs combinations (18 models). $P$: rainfall, $T_{avg}$: average air temperature, $E_p$: potential evaporation, $E_a$: actual evaporation, $S_u$: root-zone soil moisture, $S_t$: terrestrial water storage, $Q_{run}$: total runoff and $R_r$: groundwater recharge. $P$ and $T_{avg}$ are RCM-GCM simulated while the other variables come from the hydrological model.**

The hydroclimatic variables depict contrasting seasonal changes across the RCPs over the future projection periods (Figure 6 and Figure 7). The mean monthly changes are calculated as the differences between the future periods and the historical period. Figures S11-S30 illustrate the distribution of changes in the seasonal cycle of the hydroclimatic variables across RCM-GCM combinations. Overall, monthly $P$ is projected to increase between +2.9% (RCP2.6) and +136.7% (RCP8.5) on average in the dry months (November-March) and decrease between -14% (RCP8.5) and -0.6% (RCP2.6) in the wet months (May-September) over the twenty-first century. However, the reduction in rainfall over the wet months is more accentuated from the beginning to the middle of the rainy season (May-July), whereas there is an increase in rainfall intensity over the late rainy season (August-September). Consequently, monthly rainfall is projected to be more concentrated at the end of the rainy season, i.e. in August and September, as compared to the baseline period. All RCM-GCM models agree for air temperature on the increasing trend across all RCPs over the twenty-first century. There is a clear increase for $T_{avg}$ in all months with a similar pattern for each month, depicting an increase with time from 2021 to 2100 and with increasing radiative forcing level from RCP2.6 to RCP8.5. On average, $T_{avg}$ increases between +2.4% and 16.3% (i.e., +0.6 °C and +4.4 °C) across RCPs and over the twenty-first century. Similar to future projections of $T_{avg}$, monthly $E_p$ also shows a consistent increase across RCPs and for


different periods in the twenty-first century. The average changes in $E_p$ are projected from +1.1% in 2021-2050 for RCP2.6 to
+10.2% in 2071-2100 under RCP8.5 (Figure 6). The changes in simulated monthly $E_a$ follow a similar pattern to $P$, with an
average increase between +0.2% under RCP2.6 and +8.1% under RCP8.5 for $E_a$ in the dry months and a reduction of -0.2%
and -9.9%, respectively under RCP2.6 and RCP8.5, in the wet months (Figure 7).

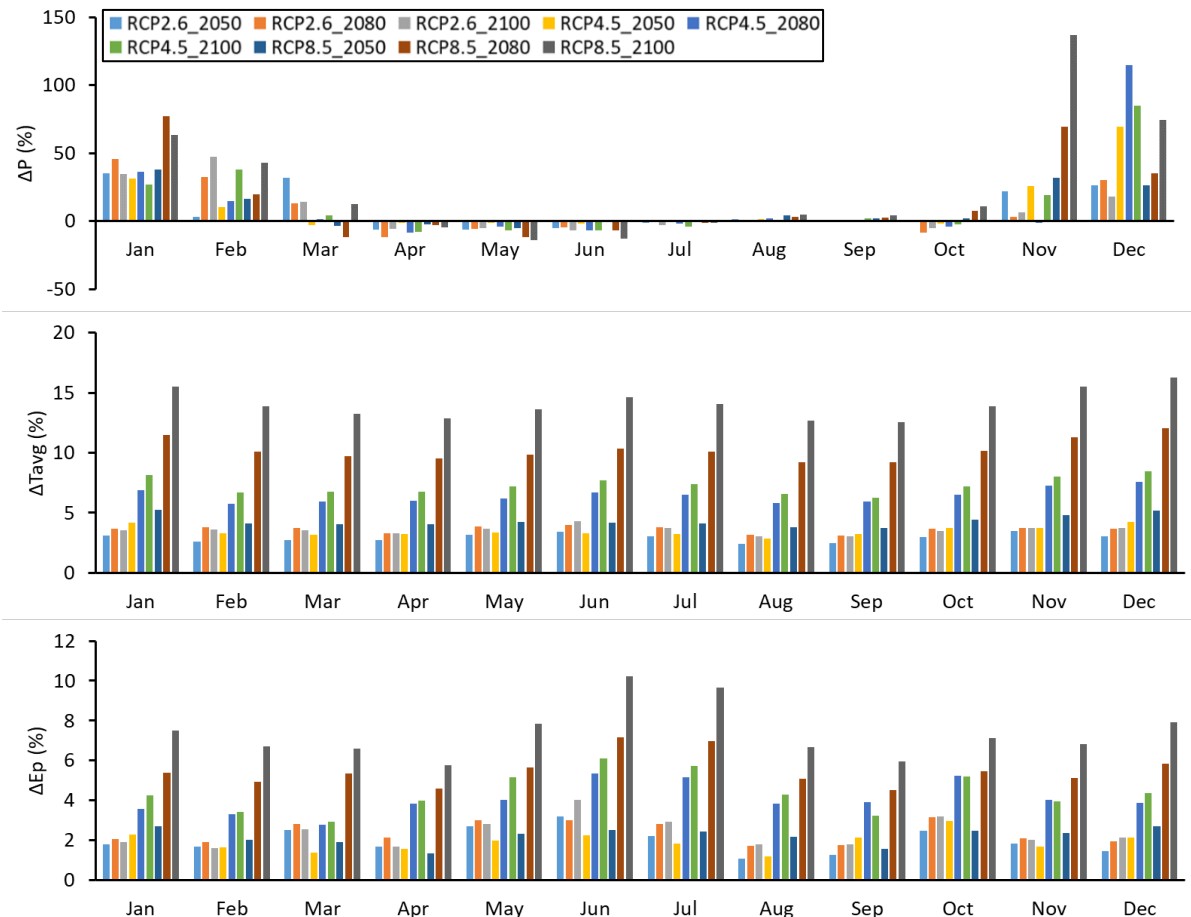

**Figure 6. Mean seasonal changes (percentage) in climatic variables ($P$, $T_{avg}$, $E_p$) over the future periods (2021-2050, 2051-2080 and 2071-2100) relative to the historical period (1991-2020).**

Projected monthly $S_u$ and monthly $S_t$ (the sum of all water stored below and above land) denote a clear decline in the future
for all RCPs except for some months in the period 2021-2050, and in the period 2071-2100 under RCP8.5 (Figure 7). The
average decrease varies between -4.1% and -0.1% for $S_u$ and between -3.6% and -0.2% for $S_t$. The monthly $Q_{run}$ and $R_r$ mainly
increase with the exception of some scenarios in some months. The projected monthly changes vary between -16.5% and
+173.9% for $Q_{run}$ and between -21.8% and +344.3% for $R_r$. The strong increases in $Q_{run}$ and $R_r$ are identified in April, when
all RCM-GCMs project a decrease in $E_a$ (Figure 7).









**Figure 7. Mean seasonal changes (percentage) in hydrological variables ($E_a$, $Q_{run}$, $R_r$, $S_u$, $S_t$) over the future periods (2021-2050, 2051-2080 and 2071-2100) relative to the historical period (1991-2020).**

Overall, the inter-model (i.e., RCM-GCMs) variability is lower for the historical period than for the future periods and increases with increasing radiative forcing level from RCP2.6 to RCP8.5 (Figure 8). Among the climatic variables, low inter-model variabilities are simulated for monthly temperatures and potential evaporation, with an average $V_2$ varying from 1% to 4% over the twenty-first century. However, $P$ shows higher inter-model variabilities with $V_2$ ranging between 30% and 44%, which have a direct repercussion on the projected hydrological variables. The highest inter-model variabilities are projected for groundwater recharge, exceeding $V_2$ of 48%. The monthly inter-model variability for all the hydroclimatic variables and all the RCPs are illustrated in Figure S31. It is noteworthy that the inter-model variability for monthly rainfall, actual evaporation, total runoff and groundwater recharge is lower during the wet months (May-September) than the dry months (November-March), while an almost constant trend across months is simulated for the other variables (Figure S31).

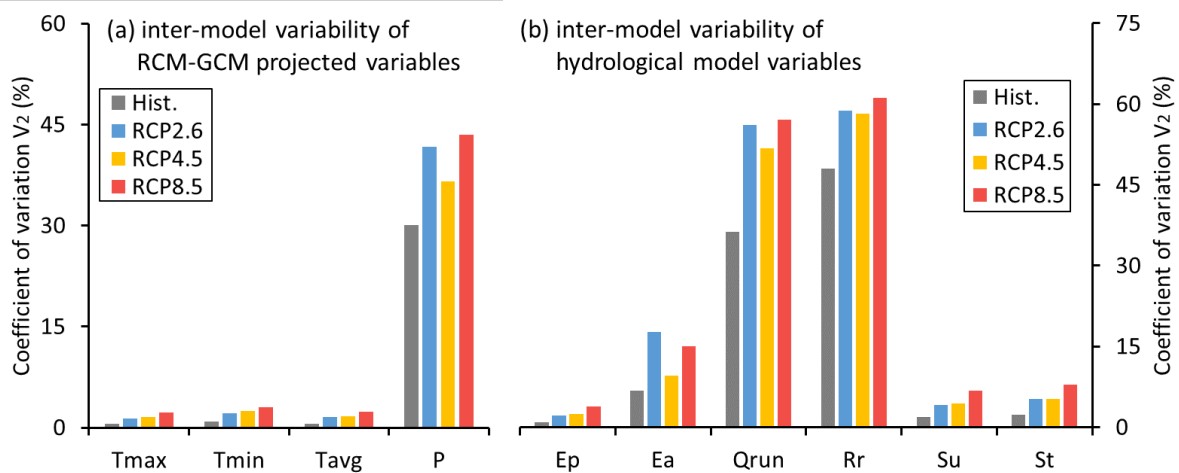

**Figure 8. Average inter-model (i.e., RCM-GCMs) variability expressed with the second-order coefficient of variation ($V_2$) for monthly hydroclimatic variables ($T_{max}$, $T_{min}$, $T_{avg}$, $P$, $E_p$, $E_a$, $Q_{run}$, $R_r$, $S_u$, and $S_t$) over the historical period (1991-2020) and the future period (2021-2100) under various RCPs.**

## 4.4 Annual changes in hydroclimatic variables and water availability

Considering all RCPs over the historical period in the VRB, the multi-model ensemble mean of long-term annual estimates of climatic variables is as follows: $T_{avg}$ = 28.4 °C, $E_p$ = 2580 mm/year, $P$ = 994 mm/year. The trends in annual estimates of projected hydroclimatic variables and water availability are illustrated in Figure S32. Water availability is assessed as the net water flux into the land surface, using a proxy metric that is the difference between rainfall and actual evaporation ($P$-$E_a$) (Konapala et al., 2020;Greve and Seneviratne, 2015;Mishra et al., 2017). The largest uncertainty in annual climate projections over the historical period is observed for rainfall with a $V_2$ of 3.2% for the inter-model variability.

The changes in the future projections of annual climatic and hydrological processes over the twenty-first century (2021-2100) as compared to the historical period (1991-2020) show contrasting trends between RCPs and between projection periods



(Figure 9). All climatic variables (rainfall, temperature and potential evaporation) are projected to increase over the twenty-first century for all RCPs and projection periods, at the exception of rainfall that is expected to decrease under RCP2.6 and RCP4.5, and increase under RCP8.5.

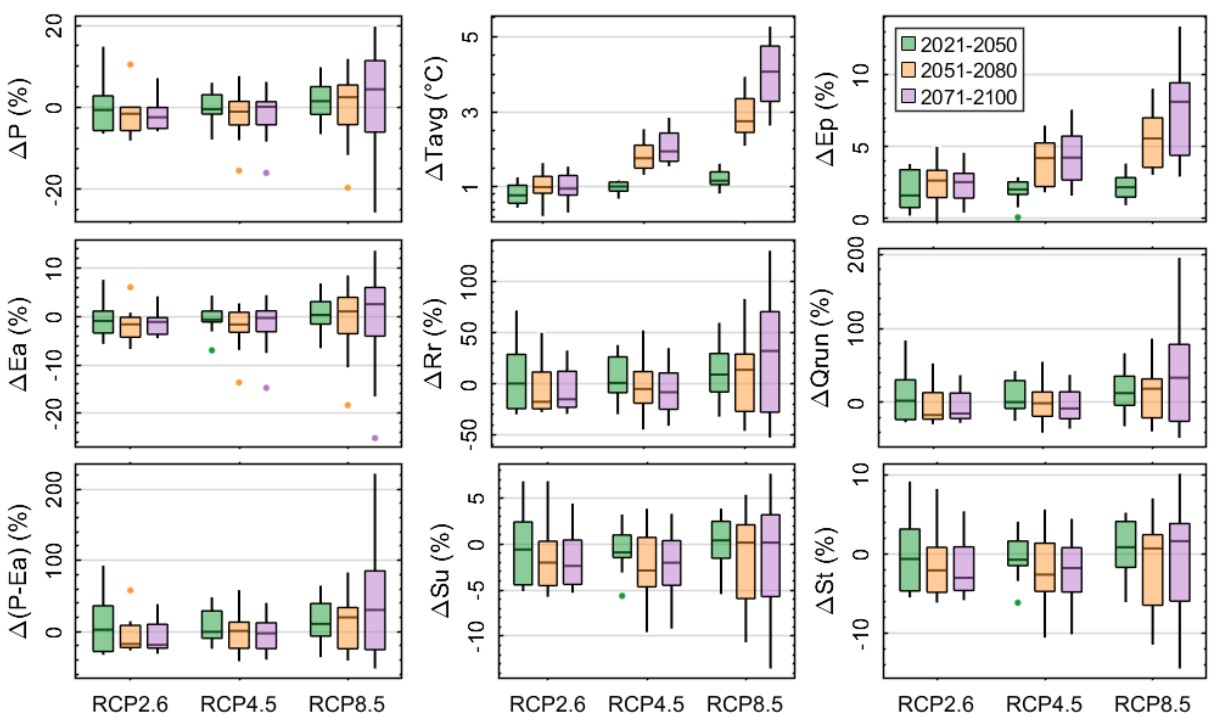

**Figure 9. Annual changes in future hydroclimatic variables and water availability ($P\text{-}E_a$) relative to the historical period for RCP2.6 (9 models), RCP4.5 (16 models) and RCP8.5 (18 models).**

The median annual changes in climatic variables over 2021-2100 are amplified with increasing radiative forcing levels, and vary between RCP2.6 ($T_{avg}$ = +1 °C, $E_p$ = +2.3%, $P$ = -1.6%), RCP4.5 ($T_{avg}$ = +1.6 °C, $E_p$ = +3.5%, $P$ = -0.5%) and RCP8.5 ($T_{avg}$ = +2.7 °C, $E_p$ = +5.3%, $P$ = +2.82%). More importantly, there is a 100% agreement among the RCM-GCM models concerning the increase in air temperature across all the RCPs and projection periods. Therefore, there is a high confidence in future climate warming of the VRB region in West Africa during the twenty-first century. These findings align with previous studies in West Africa (Dosio et al., 2020;Todzo et al., 2020;Jin et al., 2018). Similar to Dosio et al. (2019), a lower RCM-GCM agreement of 63% is found for future changes in rainfall, thereby highlighting the complexity of modelling climate in West Africa (Fitzpatrick et al., 2020;Panthou et al., 2012).

In contrast to the climatic variables, the hydrological variables (total runoff, actual evaporation, groundwater recharge, soil moisture and terrestrial water storage) decrease under RCP2.6 and RCP4.5, and only increase under the RCP8.5 scenario during the period 2021-2100 (Figure 9), thereby following the patterns of the projected changes in rainfall. The multi-model ensemble mean of long-term annual estimates of hydrological variables for all RCPs over the historical period is as follows: $E_a$ = 929 mm/year, $Q_{run}$ = 71 mm/year, $R_r$ = 11.4 mm/year, $S_u$ = 0.58 mm/mm and $S_t$ = 490 mm (Figure S32). The median





annual changes in hydrological variables over 2021-2100 vary between RCP2.6 ($E_a$ = -1.2%, $Q_{run}$ = -9.9%, $R_r$ = -10.9%, $S_u$ = -1.7%, $S_t$ = -1.9%), RCP4.5 ($E_a$ = -0.8%, $Q_{run}$ = -2.9%, $R_r$ = -4.2%, $S_u$ = -1.9%, $S_t$ = -1.7%) and RCP8.5 ($E_a$ = +1.3%, $Q_{run}$ = +21.4%, $R_r$ = +18.3%, $S_u$ = +0.3%, $S_t$ = +1.1%). It can be concluded from these findings that an intensification of the hydrological cycle (i.e., increase in the magnitude of hydrological variables) in the VRB is projected only under the RCP8.5 scenario.

The multi-model ensemble mean of water availability is 65 mm/year over the historical period. The changes in water availability are driven by the variations in rainfall, with a projected median increase of +20.3% only under RCP8.5 over the twenty-first century, and a reduction of -11.3% under RCP2.6 and -0.6% under RCP4.5.

**4.5 Spatial patterns of hydroclimatic variables across climatic zones**

   The spatial patterns of the inter-model median of annual climatic variables show a north-south gradient of increasing rainfall
and decreasing air temperature and decreasing potential evaporation (Figure 10), in line with the aridity gradient of the eco-climatic zones in the VRB (Figure 1). The spatial average of the multi-model ensemble of hydroclimatic variables per climatic zones in the VRB considering all RCPs over the historical period is given in Table 3. Annual average of hydroclimatic variables per climatic zones corresponding to the median of all RCM-GCM combinations and all RCPs over the historical period.

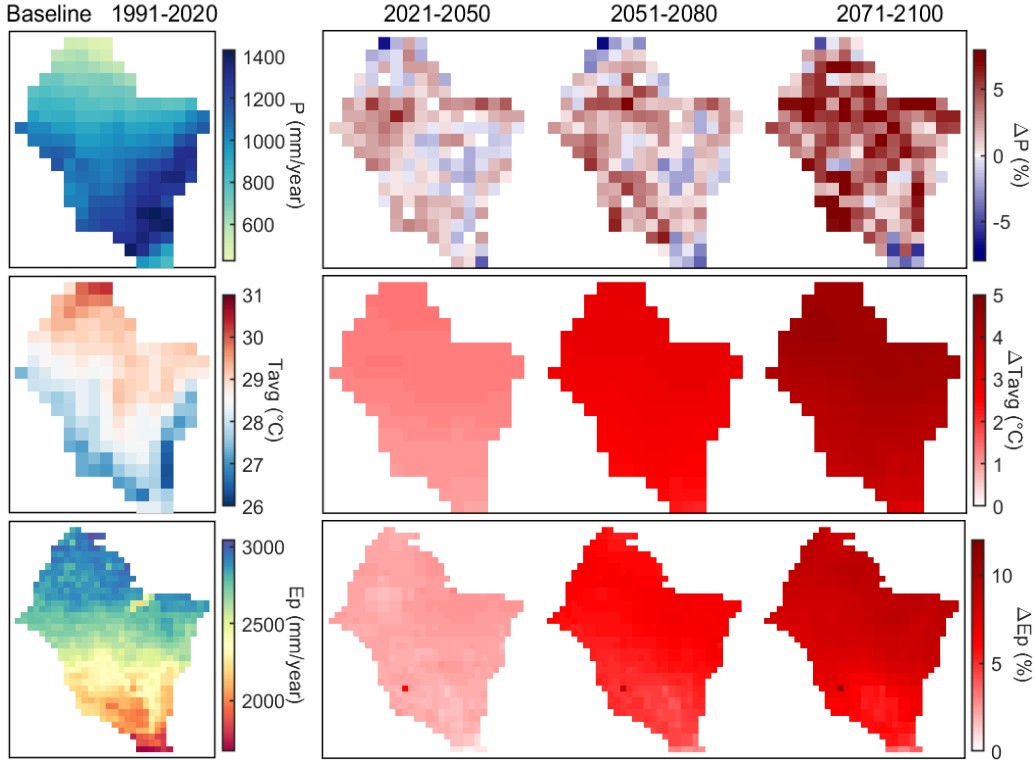

**Figure 10. Spatial patterns of the long-term average of annual climatic variables over the historical period (1991-2020) and changes over future periods (2021-2100) under RCP8.5 (median of 18 models).**





The changes in spatial patterns of climatic variables under RCP8.5 illustrate a clear increase in air temperature and potential evaporation over the twenty-first century, with higher increase rates in the Sahelian zone and the northern regions of the VRB (Figure 10, see other RCPs in Figures S33-S37). The future projections show a decrease in rainfall under RCP2.6 and RCP4.5

from the Sahelian to the Sudanian zones, which correspond to the northern and central parts of the VRB (Figures S33-S37). The increase in rainfall is projected mainly under RCP8.5, with higher increase rates localized from the Guinean to the Sudano-Sahelian zones, which correspond to the south and central regions of the VRB (Figure 10).

**Table 3. Annual average of hydroclimatic variables per climatic zones corresponding to the median of all RCM-GCM combinations and all RCPs over the historical period.**

| | | Climatic zones in the VRB | | | |
|---|---|---|---|---|---|
| | VRB | Sahelian | Sudano-Sahelian | Sudanian | Guinean |
| Aridity index ($E_p/P$) | 2.6 | 5.6 | 3.6 | 2.5 | 1.8 |
| $P$ (mm/year) | 991 | 516 | 782 | 1035 | 1243 |
| $T_{avg}$ (°C) | 28.4 | 29.7 | 29.0 | 28.3 | 27.8 |
| $E_p$ (mm/year) | 2580 | 2915 | 2849 | 2577 | 2248 |
| $E_a$ (mm/year) | 927 | 498 | 754 | 977 | 1110 |
| $Q_{run}$ (mm/year) | 69 | 17 | 27 | 57 | 132 |
| $S_u$ (mm/mm) | 0.58 | 0.32 | 0.50 | 0.59 | 0.67 |
| $S_t$ (mm) | 490 | 148 | 408 | 506 | 593 |
| $R_r$ (mm/year) | 11 | 5 | 5 | 10 | 21 |
| $P - E_a$ (mm/year) | 70 | 17 | 28 | 59 | 133 |


The spatial patterns of the hydrological variables under RCP8.5 are depicted in Figure 11 (see Figures S38-S43 for other RCPs). Changes in annual actual evaporation, surface runoff, groundwater recharge depict similar spatial patterns to those of rainfall, with the exception that surface runoff and groundwater recharge tend to increase in the Sahelian zone. Soil moisture and terrestrial water storage projections show a persistent decrease over most of the VRB area and particularly over the eastern

regions. The analysis of projected future water availability per climatic zones in the VRB reveals an opposite pattern to the "dry gets drier, wet gets wetter" paradigm (Byrne and O'Gorman, 2015;Greve et al., 2014). The dry regions of the VRB (Sahelian and Soudano-Sahelian zones) are projected to become wetter, while the wet (Sudanian and Guinean zones) are projected to become drier under RCP2.6 and RCP4.5. However, all climatic zones in the VRB are projected to become wetter under RCP8.5.

The spatial inter-model variability depicts different patterns across hydroclimatic variables and across RCPs and projection periods (Figures S44-S54). For RCP8.5, the highest differences among the simulations from climatic models are found generally in the northern dry regions (Sahelian zone) and in the forested southwestern wet region (Guinean zone) of the VRB (Figure 12).








**Figure 11. Spatial patterns of the long-term average of annual hydrological variables and water availability over the historical period (1991-2020) and changes over future periods (2021-2100) under RCP8.5 (median of 18 models).**









**Figure 12. Spatial patterns of inter-model variability of hydroclimatic variables under RCP8.5 (18 models).** *P*: rainfall, $T_{avg}$: average
air temperature, $E_p$: potential evaporation, $E_a$: actual evaporation, $S_u$: root-zone soil moisture, $S_t$: terrestrial water storage, $Q_{run}$:
total runoff and $R_r$: groundwater recharge.

### 4.6 Change in high and low flows

The streamflow ($Q$) projections at the outlets of the sub-basins in the VRB are used for the analysis of high and low flows.
Here, high flows correspond to streamflow with a return period of 10 years, i.e. streamflow peaks that are equalled or exceeded
for 10 percent of the time ($Q_{10}$). Low flows represent streamflow that is equalled or exceeded for 90 percent of the time ($Q_{90}$).
Therefore, $Q_{90}$ and $Q_{10}$ are the tenth and ninetieth percentiles of daily streamflow, corresponding to the 90% and 10%
probability of exceedance, respectively. An increase in $Q_{10}$ implies an increase in flood risk, whereas a decrease in $Q_{90}$
represents a higher risk for river drought (Aich et al., 2014). The streamflow gauges are Bui-Amont, Daboya and Saboba at
the outlets of the Black Volta, White Volta and Oti sub-basins (Figure 1), respectively. No gauge at the outlet of the Lower
Volta sub-basin was available for the study.

The historical and future projections of high and low flows are illustrated in Figure S55. The projected future changes in high
and low flows are summarized in Figure 13. Changes in low flows indicate an overall median increase in $Q_{90}$ in all the sub-
basins under RCP8.5 as follows: Black Volta (+30%), White Volta (+16%) and Oti (+14%), with an RCM-GCM agreement
on the direction of change of 69%, 63% and 57%, respectively. It is noteworthy that median $Q_{90}$ is projected to decrease by -
4% in the White Volta over the period 2051-2080 under RCP8.5. The trends in the median changes in $Q_{90}$ are contrasted across
the future periods for RCP2.6 and RCP4.5. Table S1-S2 provide details on the direction of change between the RCM-GCM
combinations, which in average varies between 50% and 78% for $Q_{90}$, and between 56% and 75% for $Q_{10}$, depending on RCPs
and climate projection periods.

For high flows, $Q_{10}$ is projected to increase in all the sub-basins under RCP8.5 on average by +20% in the Black Volta, +2%
in the White Volta and + 6% in the Oti over the period 2021-2100, with an RCM-GCM agreement on the direction of change
of 65%, 63% and 59%, respectively. However, it is noteworthy that $Q_{10}$ for the period 2051-2080 under RCP8.5 is projected
to decrease by -14% in the White Volta and by -10% in the Oti. Under RCP2.6 and RCP4.5, a decrease in $Q_{10}$ is projected
from 2051 to 2100 in all the sub-basins. The highest median decrease in $Q_{10}$ over the twenty-first century in the sub-basins is
-6% in the Black Volta under RCP4.5, -16% in the White Volta under RCP2.6 and -20% in the Oti under RCP2.6. Both high
and low flows are projected to increase with different magnitudes under all RCPs in the Black Volta and the White Volta over
the period 2021-2050, while low flows are projected to increase under all RCPs in the Oti over the period 2051-2080.

Among the sub-basins, the Black Volta presents the highest probability of an increase in the 10-year flood over the twenty-
first century. The White Volta is prone to an increase in flood risk in the period 2021-2050, while river drought risk is projected
to increase over the period 2051-2100.



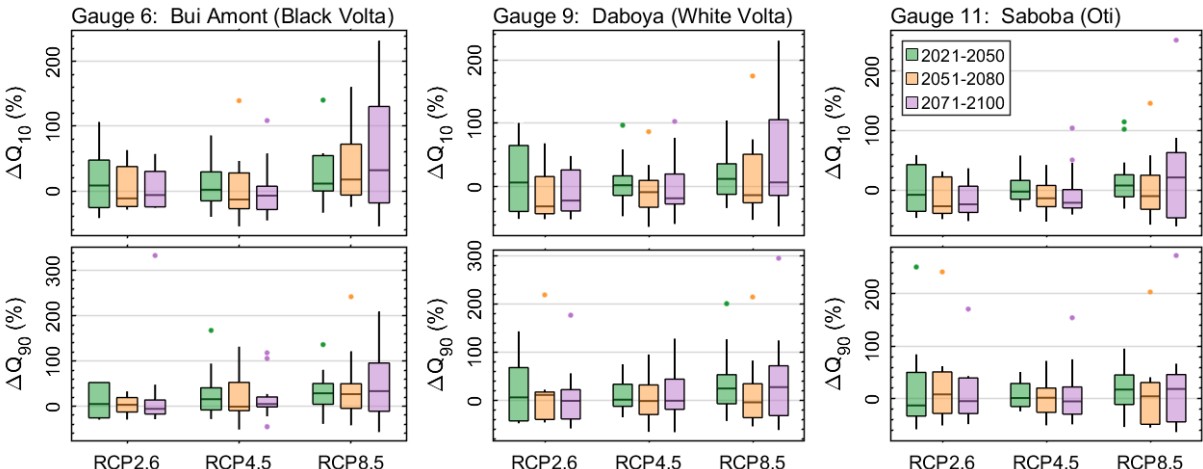

**Figure 13. Changes in future high flows ($Q_{10}$) and low flows ($Q_{90}$) in the major sub-basins of the VRB (Black Volta, White Volta, Oti). The changes in percentage over the future periods are relative to the historical period (1991-2021).**

## 4.7 Climate sensitivity

The sensitivity of selected hydrological variables to changes in climatic variables over the historical and future periods is illustrated in

Figure 14 (see Figure S56 for additional variables). The changes in hydroclimatic variables are estimated in percentage over the three future periods compared to the historical period. Larger changes in hydroclimatic variables are simulated with increasing radiative forcing levels over the future period.

The VRB illustrates highly nonlinear behavior. An increase in annual rainfall by 10% results in an increase in total runoff by ~50%, and an increase in actual evaporation by ~6%. A 10% decrease in annual rainfall leads to ~30% decrease in total runoff, and ~7% decrease in actual evaporation. Annual actual evaporation represents 93% of annual rainfall, while total runoff represents 7% of annual rainfall. However, small changes in the rainfall regime might strongly affect the total runoff regime. A 5% increase in annual average air temperature leads to an increase of ~3% in potential evaporation. However, heteroscedasticity is observed for changes in air temperature and potential evaporation as the variability increases with higher rates of change (

Figure 14f). This analysis reveals a high sensitivity of major fluxes ($Q_{run}$ and $E_a$) to even small changes in $P$ in the VRB, a finding consistent with previous studies in the region (Roudier et al., 2014), and more generally in dryland regions globally (Berghuijs et al., 2017).







**Figure 14. Climate sensitivity of hydrological processes in the VRB over the historical period ((a), (c) and (e)) with associated future changes ((b), (d) and (f)). The coloured dots represent RCM-GCM combinations per RCP and the colour dashed lines represent the fitted linear regression.**

## 5 Discussion

A direct comparison of our results to previous studies in the region is restricted by the differences in the choice of RCM-GCM models, RCPs, future projections periods, and the baseline period, which alone might lead to different outcomes (Liersch et al., 2020). However, our results generally corroborate with comparable previous studies in the region (Aich et al., 2016;Roudier et al., 2014;Sidibe et al., 2020). It is noteworthy that total runoff is projected to increase while rainfall decreases under the RCP2.6 and RCP4.5 during the period 2021-2050. This paradoxical phenomenon of rainfall-runoff negative correlation is commonly referred to as the "Sahelian paradox" (Mahé and Paturel, 2009). The percentage of agreement of 63% for hydrological projections between the climate models is similar to that of rainfall, which supports that rainfall is the key driver of hydrological processes in the VRB, thereby confirming the findings of Roudier et al. (2014). Vetter et al. (2015) also found a high variability in rainfall projections among climate models in the Upper Niger basin. Therefore, the improvement of rainfall representation in climate models, which hardly represent the West African monsoon (Akinsanola et al., 2020;Philippon et al., 2010;Xue et al., 2010), would ultimately enhance the reliability of the assessment of climate change impacts on water resources (Dosio et al., 2021). The development of convection-permitting climate models to better capture complex climate features in the region at higher resolutions would help (Berthou et al., 2019;Kendon et al., 2017;Kendon et al., 2019). The projected decrease of rainfall under RCP2.6 and RCP4.5, whereas an increase is projected under RCP8.5, can be explained by the higher warming level under RCP8.5. Higher warming would lead to increased evaporation, resulting in increased water vapour in the atmosphere, which would turn into increased precipitation (Donat et al., 2016;Trenberth, 2011).

The large ensemble of RCM and GCM datasets used in this study allows the quantification of model-related projection uncertainties in terms of inter-model variabilities. In general, the selection of the best-performing climate models for hydrological modelling is not straightforward (Dosio et al., 2019;Hakala et al., 2019). Therefore, all the RCM-GCM combinations are used in this study without weighting or excluding models because the performance of models in the future is not necessarily related to their performance in the present (Dosio et al., 2020). More detailed analyses using methods for climate model selection (Kiesel et al., 2020;Ross and Najjar, 2019;Merrifield et al., 2020;Abramowitz et al., 2019;Ahmed et al., 2019) with the updated CMIP6 models (Eyring et al., 2016) and the new Shared Socioeconomic Pathways (SSPs) (O'Neill et al., 2017;O'Neill et al., 2014;Riahi et al., 2017) are left for future work. For uncertainty dependence on ensemble composition, further attention should be given to single-model initial condition large ensembles (SMILEs) (Milinski et al., 2020;Maher et al., 2021).

A further potential limitation arises from the imposed stationarity of the dependences between variables related to the used bias-correction method, although the used R2D2 method assumes some non-stationarity in the marginals (i.e., univariate distributions). However, predicting non-stationarity of dependence biases under climate change is not straightforward. With





advances in multivariate bias correction methods (François et al., 2020), the added value of methods that consider non-stationarity in climate (e.g., dOTC; Robin et al., 2019) could be investigated in climate impact studies on water resources (Yang et al., 2021b).

Although multiple RCMs, GCMs and RCPs are used in this study, a single hydrological model is used for the hydrological projections. Therefore, the results are also subject to potential deficiencies of the mHM model, as hydrological models are known to be a source of uncertainty in climate change impact studies (Vetter et al., 2017;Hattermann et al., 2018;Mendoza et al., 2015;Giuntoli et al., 2015;Hagemann et al., 2013). For instance, mHM does not explicitly consider the effect of vegetation dynamics associated with climate change that would modify runoff processes, which can be a limitation that is also observed in previous studies (Duethmann et al., 2020;Hanus et al., 2021;Wu et al., 2016). However, the mHM model used in this study has been thoroughly calibrated to provide realistic simulations of hydrological state variables and fluxes in the VRB (Dembélé et al., 2020a).

In addition to the hydrological model itself, the choice of the method for the calculation of potential evaporation can introduce uncertainties in the hydrological projections as reported in previous studies (Prudhomme and Williamson, 2013;Seiller and Anctil, 2016). The lack of atmospheric coupling between purely temperature driven potential evaporation and the hydrological cycle in the hydrological model projections means some important feedbacks, such as surface resistance, may be missing (Milly and Dunne, 2016;Yang et al., 2019), and will not incorporate other mechanisms important in atmospheric demand. However, it is worth noting that the importance of this missing feedback has thus far only been evaluated for conditions in which water availability is not a limiting factor, a condition that is rarely met in the VRB. Moreover, even a more suitable method for estimating potential evaporation (e.g. Penman-Monteith), which can also be adjusted for surface resistance changes in future climates (Yang et al., 2019), is to the first order still expected to generate potential evaporation rates higher than precipitation. Therefore, the increase in soil moisture and actual evaporation will likely still be very large in response to the increase in precipitation. Nonetheless, the long-term $CO_2$ – vegetation driven evaporation feedbacks not considered here could be very important for lower flow and drought conditions and need consideration in future work. The use of RCMs to downscale, which generally do not have the appropriate $CO_2$ – vegetation feedbacks in contrast to GCMs, is also an issue that needs to be addressed in future downscaling projects.

Furthermore, the hydrological projections generated in this study focus on changes in climate and do not explicitly account for land use and land cover change or changes in water management practices in the VRB. Although, land use and land cover changes play an important role in the hydrological processes, the primary focus in this study is constraining the impact of climate change alone. However, land use changes are assumed to be accounted for to some extent in the RCPs, as their development is based on assumptions regarding future evolution of land use and land cover (Van Vuuren et al., 2011). Nevertheless, given the dominance of the evaporation response, future work accounting for potential land use and land cover changes in response to the climate scenarios provided here will be needed. Finally, large basins studies of climate change impacts on water resources need to consider human interactions with the hydrological cycle to better integrate the co-evolution of the human–water systems across scales (Yang et al., 2021a).





## 6 Conclusion

A large ensemble of twelve GCMs from CMIP5 and five RCMs from CORDEX-Africa is used to investigate the impacts of climate change on water resources in the Volta River basin under three RCPs. The climate projection datasets are used to force the fully distributed mesoscale Hydrologic Model (mHM) over the twenty first century. Changes in hydrological processes over the future periods 2021-2050, 2051-2080 and 2071-2100 are estimated relatively to the historical period 1991-2020. The results reveal contrasting changes in the hydrological cycle depending on RCPs and future projection periods. The key findings are summarized as follows:

- Climate warming is projected in the Volta basin as all RCM-GCM projections predict an increase in air temperature under all RCPs, accompanied by an increase in annual potential evaporation.
- Rainfall is projected to decrease under RCP2.6 and RCP4.5, while an increase is projected under RCP8.5, with a direct correlated and contrasting impact on water availability in the Volta River basin. Compared to temperature, there are more uncertainties in the trend of the changes in rainfall projections as there is only an agreement of 63% on the direction of change between the RCM-GCM models, which leads to more uncertainty in the prediction of hydrological variables.
- The seasonality of rainfall is projected to shift forward in the future, with the concentration period of the rainy season moving towards the months of August and September.
- Annual actual evaporation, total runoff, groundwater recharge, soil moisture and terrestrial water storage decline under RCP2.6 and RCP4.5, while they increase under RCP8.5 following the trends in rainfall. In fact, a strong sensitivity of hydrological processes to climate variability is found.
- The analysis of high and low flows suggests an increased risk for floods under RCP8.5 over the twenty-first century, while an increased risk for hydrological drought is projected under RCP2.6 and RCP4.5 from the mid-twenty-first century.
- The spatial projections of future water availability per climatic zones depict a "dry gets wet and wet gets dry" pattern under RCP2.6 and RCP4.5.
- Contrary to the other RCPs, under RCP 8.5, the projected climate changes lead to a clear intensification of the entire hydrological cycle, i.e. an increase in the magnitude of hydroclimatic variables.

The changes in the hydrological cycle have important implications for future floods and droughts in the Volta basin, thereby amplifying the vulnerability of the local population to climate change. These findings can contribute to the elaboration of regional adaptation and mitigation strategies of climate change. However, significant inter-model variabilities of the climate models and low to moderate agreement between RCM-GCM combinations on the direction of changes highlight the complexity and uncertainties related to the assessment of climate change impacts on water resources. Therefore, more work is required to improve climate modelling in West Africa. A strong collaboration between climate and water resources scientists, practitioners and policymakers is key for advancing knowledge and development.



**Code availability.** The source code of the mHM model is available at https://doi.org/10.5281/zenodo.1299584.

**Data availability.** CORDEX data can be accessed from the ESGF database at https://esgf-data.dkrz.de, (last accessed on 22.03.2020). The hydrological modelling database is accessible at https://doi.org/10.5281/zenodo.3531873.

**Supplement.** The supplement related to this article is available online at: _to be provided by the journal_

**Author contribution.** MD conducted the analyses and wrote the manuscript. MV provided guidance for the use of the R2D2 method. All authors reviewed the manuscript and contributed to the writing.

**Competing interests.** The authors declare that they have no conflict of interest.

**Acknowledgments.** We are grateful to the developers of mHM at CHS/UFZ (Germany) for their open-source model. We thank the providers of the streamflow data obtained from the Volta Basin Authority (VBA), the Direction Générale des
Ressources en Eau (DGRE) of Burkina Faso, the Hydrological Services Department (HSD) of Ghana and the Direction Générale de l'Eau et de l'Assainissement (DGEA) of Togo. We thank the developers of the global and regional climate models participating to the CORDEX initiative.

**Financial support.** Moctar Dembélé was supported by the Swiss Government Excellence Scholarship (2016.0533/Burkina
Faso/OP), the Doc.Mobility fellowship (SNF, P1LAP2_178071) of the Swiss National Science Foundation, and the Hydro-JULES visiting scientist fellowship (UKCEH; NERC NE/S017380/1). Bettina Schaefli was supported by a research grant of the Swiss National Science Foundation (SNF, PP00P2_157611). Mathieu Vrac has been supported by the CoCliServ project, which is part of ERA4CS, an ERA-NET initiated by JPI Climate and co-funded by the European Union (grant no. 690462).

**Review statement.** This paper was edited by _Editor's name_ and reviewed by _referees names(or anonymous)_.

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
