# Peer review of "Contrasting changes in hydrological processes of the Volta River basin under global warming"

_Hydrology and Earth System Sciences, 2021_

## Author Comment (AC1)

**Reply to Anonymous Referee #1 on the manuscript**
**"Contrasting dynamics of hydrological processes in the Volta River basin under global warming"**
**(Manuscript hess-2021-525)**
**by M. Dembélé et al.**

I enjoyed reading this manuscript. The authors used a distributed model for assessing climate change impacts on different fluxes and discharge output in Volta basin. The ms is quite elaborated and fits well with HESS standards. I only have several concerns regarding "dynamics" and "uncertainty" results presented in the manuscript. Moreover, climate gradient in the basin seems to ruin (dominate) AET patterns censoring vegetation dynamics.

We thank the reviewer for this very positive overall appreciation of our work and the constructive review and valuable comments. Below are our responses to each of the comments.

Specific comments:

-Title is very catch but I couldn't find much on "dynamics" presented in the results except for the Fig5, 13 and 14.

The term "dynamics" is used here to refer to the temporal and spatial changes observed in the hydrological processes as a result of climate change. In addition to Figures 5, 13 and 14, Figures 6, 7, 9 show the temporal dynamics and Figures 10 and 11 show the spatial dynamics of hydrological variables. Moreover, additional figures are available in the supplementary material to support our analyses on the dynamics of hydrological variables (e.g. Figure S1 to Figure S30). However, we will do further analyses on the seasonality as suggested by the reviewer below.

-In addition to Fig13, the readers would be curious to see Lahaa and Blösch (2006) type seasonality figures for regime (a kind of dynamics) changes/shifts in the basin. The seasonality indices could be adopted to low and high flows as done in different other papers below. Event definition is key here for counting them i.e. Q95 and Q5 for low and high flows.

-Seasonality shift is only mentioned for rainfall at line 491 in conclusions but there is room for assessing shifts in high and low flow occurrence (dynamics) and seasonality.

Figures 4-6-9 in Laaha and Blösch (2006) DOI: 10.1002/hyp.6161 are good examples.

Similar applications in climate research:

https://mdpi-res.com/d_attachment/water/water-12-03575/article_deploy/water-12-03575-v3.pdf

https://mdpi-res.com/d_attachment/water/water-11-00925/article_deploy/water-11-00925-v2.pdf

We thank the reviewer for sharing approaches to investigate shift in high and low flows. We will investigate the method of circular statistics adopted in the suggested papers to assess changes in the days of occurrence of low and high flows and their variability, which is an indicator of seasonality.

-Fig 11, 12, S38, S49 (AET in particular) are mostly dominated by climate gradient and not showing vegetation dynamics. The authors should find a way to exclude the dominant effect of rainfall using a normalization procedure. A new procedure is proposed in this paper

Example:

https://www.preprints.org/manuscript/202111.0225/v1

However, there must be other methods approaches in the literature for deblurred AET pattern maps by removing climate gradient.

https://www.researchgate.net/publication/338208138_Image_Deblurring_Techniques_-A_Detail_Review

We agree with the reviewer that actual evaporation (Ea) patterns are governed by rainfall patterns in the Volta basin. The study area is located in a semi-arid zone where the main driver of the water cycle is rainfall, with annual Ea accounting for 80% of rainfall. Therefore, Ea naturally follows the patterns of rainfall, which can also be observed for vegetation, i.e. there is more vegetation in the southern part of the basin where rainfall is high and less as we move towards the north (Figure 1). The normalization of Ea would not bring substantial information to the reader as we are interested in analyzing the main hydrological variables and not their derivatives (e.g. the evaporative index, calculated as the ratio of Ea to P). We understand the potential value of the normalization for model calibration but the goal of our manuscript is not focused on calibration, which we did in a previous study (https://doi.org/10.1029/2019wr026085)

-Line 196: "Uncertainties in the model inputs and outputs are assessed in terms of variability be"

Indicating V2 estimation (or even COV coefficient of variation) as uncertainty assessment is quite ambitious without a systematic uncertainty propagation like in GLUE by Keith Beven.

In short, this vague sentence should be revised as smth like "variability in the model inputs and outputs are assessed using V2 statistics".

The sentence will be revised to avoid confusion.

---

## Author Comment (AC2)

**Reply to Anonymous Referee #2 on the manuscript**
**"Contrasting dynamics of hydrological processes in the Volta River basin under global warming"**
**(Manuscript hess-2021-525)**
**by M. Dembélé et al.**

Dembele et al. (2021) evaluate uncertainty in hydrologic variables based on twelve GCMs from CMIP5 dynamically downscaled by five RCMs over the Volta River Basin in Africa. This paper is written clearly, and it is an interesting study, particularly considering the expected population doubling between 2010 and 2050 and implications, which can be associated with changes in water redistribution. I have some minor comments which I kindly ask the authors to consider during manuscript revisions.

We thank the reviewer for this very positive overall appreciation of our work and the constructive review and valuable comments. Below are our responses to each of the comments.

Some less critical analyses could possibly be moved to Supplement to ease the reading and to keep the focus on the most important results. Mainly, if a figure is explained in one sentence, it requires to be moved to Supplement, e.g. Fig.12). Possibly, the Supplement itself is also very lengthy, and I suggest, remove less important figures.

We agree with the reviewer to move Fig.12 to the supplementary material. Moreover, we will reduce the content of the supplementary material.

On the other hand, I have missed some evaluation of hydrologic model performance. I understand that was done in earlier studies, but it might be helpful to include observation-based climatology, say from ERA5, into Figure 5.

The paper is already long as the reviewer highlighted in the previous comment. Therefore, we prefer to refer the reader to the full study on model calibration where model performance is detailed.

For consistency, adding ERA5 climatology to Figure 5 would require also adding the climatology of the other reference datasets used for bias-correction (i.e. PGF v3, WFDEI, EWEMBI, MERRA-2, JRA-55 TAMSAT v3.0, CHIRPS v2.0, ARC v2.0, MSWEP v2.2, PERSIANN-CDR v1r1), which would result in a glutted figure. We think Figure 3 already provides a good comparison of the satellite and reanalysis datasets to the RCM-GCM datasets.

By using the historical period 1991-2020, the historical and RCP simulations get mixed. Should not be the historical period be considered only prior 2005?

The historical period can be chosen beyond 2005 as done in some previous studies (e.g. Almazroui et al. 2021; Hanus et al. 2021; Mengistu et al. 2021; Abubakari et al. 2019). As said by Hawkins and Sutton (2016): "A number of factors enter the decision about an appropriate observational reference period, for example, to be representative of the most recent conditions but long enough not to be overly influenced by random fluctuations, to be a period the public can relate to...". According to Liersch et al. (2020): "The Intergovernmental Panel on Climate Change (IPCC) will use the years 1995–2014 in its Sixth Assessment Report". The most current and widely used reference period for climate analyses by the World Meteorological Organization (WMO) is 1981–2010. We have chosen

the period 1991-2020 to have a more recent context for understanding climate change. We will further discuss the choice of our reference period in the manuscript.

The title requires changes. Please replace mainly these two words that do not fit the current version of the paper: "dynamics" and "processes".

The term "dynamics" is used here to refer to the temporal and spatial changes observed in the hydrological processes (i.e. rainfall, evaporation, runoff, groundwater recharge, soil moisture and terrestrial water storage) as a result of climate change. As suggested by the Anonymous reviewer #1, we will investigate the method of circular statistics to assess changes in the days of occurrence of low and high flows and their variability, which is an indicator of seasonality, to further consider the dynamics of hydrological processes (see https://www.nwrfc.noaa.gov/info/water_cycle/hydrology.cgi).

Which PET method was used in the climate projections? This study (e.g. https://www.nature.com/articles/nature11575) suggests that different results can be obtained for different PET methods). Could you please clarify?

We agree that different PET methods would lead to different results. In fact, we have thoroughly discussed the limitations of the choice of PET methods at lines 454-467 but we missed to mention the method we have used (i.e. Hargreaves and Samani). This will be corrected in the new version of the manuscript.

Bootstrapping should be considered for the analysis presented in Fig. 9, Fig. 13, to account for varying sample sizes between RCPs.

We understand the suggestion of bootstrapping as we do not have the same number of models for each RCP. The bootstrapping can help to randomly select a common number of RCMs to do the analysis and repeat it. However, we think it will not solve the problem, as the underlying distributions will still not be the same. One solution would be to restrict these analyses to the models for which we have the 3 RCPs available but it would restrict our analysis to only 5 RCM_GCM runs. Therefore, we prefer keeping the results as it with as many RCM_GCMs as possible. However, we will highlight the issue of varying sample sizes between RCPs in the discussion.
* * *
Textual suggestions:

lines 16-17: Rephrase abstract, the first sentence, into something like: "This study conducts a comprehensive evaluation of the impacts of climate change on the West Africa Volta River basin's water resources, as the region is expected to be hardest hit by global warming."

Ok.

lines 22-23: Reformulate into something like: "The bias-corrected climate projections are then used as input to the mesoscale Hydrologic Model (mHM) for hydrological projections over the twenty-first century (1991-2100)."

Ok.

Lines 31-32: rephrase into: "and amplifying the local population's vulnerability."

Ok.

line 37: "at a faster rate" => "faster"

Ok.

lines 43-44: "Climate change and anthropogenic pressures increase water resources' stress (Sood et al., 2013)"

Ok.

line 45: "for" => "to"

Ok.

line 60: rephrase into "usually focused"

Ok.

line 67: reformulate to "the repercussions"

Ok.

line 72 remove "provide knowledge to"

Ok.

line 75: maybe "central" instead of "major"

"major" is more appropriate here.

Figure 1: it would be more helpful to split the grand legend block into figure panels, where individual classes are shown.

Not sure if we understood the suggestion. However, the current figure arrangement allows saving space and avoid not having many figures only for the description of the study area.

Line 94: "the assessment of" into "assessing"

Ok.

Line 123-124: Possibly rephrase into "As the RCMs downscale not all GCMs,…"

Ok. Would be "As the RCMs do not downscale all the GCMs…".

Line 188: "a steady-state"

Ok.

Figure 3 caption: Write clearly this is the historical period (keep consistency with the other figures)

Ok.

Figure 5 caption: should be: "… for the historical and future periods (under RCP8.5)."

Ok.

Figure 6-7 caption: synchronize legend (e.g., *_2050) with figure caption (2021-2050).

Ok.

**References**

Abubakari, S., Dong, X., Su, B., Hu, X., Liu, J., Li, Y., ... & Xu, S. (2019). Modelling streamflow response to climate change in data-scarce White Volta River basin of West Africa using a semi-distributed hydrologic model. Journal of Water and Climate Change, 10(4), 907-930. https://doi.org/10.2166/wcc.2018.193

Almazroui, M., Ashfaq, M., Islam, M. N., Rashid, I. U., Kamil, S., Abid, M. A., ... & Sylla, M. B. (2021). Assessment of CMIP6 Performance and Projected Temperature and Precipitation Changes Over South America. Earth Systems and Environment, 1-29. https://doi.org/10.1007/s41748-021-00233-6

Hanus, S., Hrachowitz, M., Zekollari, H., Schoups, G., Vizcaino, M., and Kaitna, R. (2021). Future changes in annual, seasonal and monthly runoff signatures in contrasting Alpine catchments in Austria, Hydrology and Earth System Sciences, 25, 3429-3453, 645 https://doi.org/10.5194/hess-25-3429-2021

Hawkins, E., & Sutton, R. (2016). Connecting climate model projections of global temperature change with the real world. Bulletin of the American Meteorological Society, 97(6), 963-980. https://doi.org/10.1175/BAMS-D-14-00154.1

Liersch, S., Drews, M., Pilz, T., Salack, S., Sietz, D., Aich, V., ... & Hattermann, F. F. (2020). One simulation, different conclusions—the baseline period makes the difference!. Environmental Research Letters, 15(10), 104014. https://iopscience.iop.org/article/10.1088/1748-9326/aba3d7

Mengistu, D., Bewket, W., Dosio, A., & Panitz, H. J. (2021). Climate change impacts on water resources in the Upper Blue Nile (Abay) River Basin, Ethiopia. Journal of Hydrology, 592, 125614. https://doi.org/10.1016/j.jhydrol.2020.125614

---

## Author Response (AR1)

**Reply to the Editor and the Referees of the manuscript**
**"Contrasting dynamics of hydrological processes in the Volta River basin under global warming"**
**(Manuscript hess-2021-525)**
**by M. Dembélé et al.**

**Reply to the Editor**

Comments to the author:

Dear Moctar Dembélé et al.,

Thank you for responding to the reviews. You have responded to the comments appropriately.

Both reviewers commented on the wording 'dynamics' used in the title. I agree that 'dynamics' is a catchy word but it does not really reflect the content of the manuscript. You stated in the paper "the contrasting changes in the hydrological cycle", which is a more appropriate phrasing then "Contrasting dynamics". When the latter is used, readers may expect to see more physical changes in the processes although I understand you use this word in a broader context to refer to changes. It can be misleading to readers who would expect to see changes in the underlying physical processes. What the manuscript contains is the quantification of hydrological impacts in the Volta River Basin under global warming. I suggest you consider rephrasing the title.

It is overall a well researched and presented manuscript. I look forward to receiving your revised manuscript.

Sincerely,

Yi He, HESS Editor

We thank the editor for handling our manuscript and for the very positive overall appreciation of our work. We agree to substitute the term "dynamics" with "changes" in the title and where appropriate in the manuscript.

New sections are added to the manuscript to provide information on the seasonality of the date of occurrence of high and low flows and their evolutions under climate change using the method of circular statistics. These are sections "3.4.3 Timing of high and low flows" and "4.6 Changes in high and low flows". More details are provided below in our answers to the referees' comments.

**Reply to Anonymous Referee #1**

I enjoyed reading this manuscript. The authors used a distributed model for assessing climate change impacts on different fluxes and discharge output in Volta basin. The ms is quite elaborated and fits well with HESS standards. I only have several concerns regarding "dynamics" and "uncertainty" results presented in the manuscript. Moreover, climate gradient in the basin seems to ruin (dominate) AET patterns censoring vegetation dynamics.

We thank the reviewer for this very positive overall appreciation of our work and the constructive review and valuable comments. Below are our responses to each of the comments.

Specific comments:

-Title is very catch but I couldn't find much on "dynamics" presented in the results except for the Fig5, 13 and 14.

We agree to substitute the term "dynamics" to "changes" in the title and where appropriate in the manuscript. Moreover, we have now done further analyses on the seasonality using the method of circular statistics as suggested by the reviewer below.

-In addition to Fig13, the readers would be curious to see Lahaa and Blösch (2006) type seasonality figures for regime (a kind of dynamics) changes/shifts in the basin. The seasonality indices could be adopted to low and high flows as done in different other papers below. Event definition is key here for counting them i.e. Q95 and Q5 for low and high flows.

-Seasonality shift is only mentioned for rainfall at line 491 in conclusions but there is room for assessing shifts in high and low flow occurrence (dynamics) and seasonality.

Figures 4-6-9 in Laaha and Blösch (2006) DOI: 10.1002/hyp.6161 are good examples.

Similar applications in climate research:

https://mdpi-res.com/d_attachment/water/water-12-03575/article_deploy/water-12-03575-v3.pdf

https://mdpi-res.com/d_attachment/water/water-11-00925/article_deploy/water-11-00925-v2.pdf

We thank the reviewer for sharing approaches to investigate shift in high and low flows. We have now used the method of circular statistics adopted in the suggested papers to assess changes in the dates of occurrence of low and high flows and their variability, which is an indicator of seasonality.

The following is added to the manuscript:

**3.4.3 Timing of high and low flows**

The timing of high and low flows is assessed by first estimating the dates on which the annual $Q_{10}$ and $Q_{90}$ occurred for each of the individual 30-year historical period and future periods. Subsequently, the method of circular statistics (Mardia, 1972, 1975) is used to calculate the mean date of occurrence (measure of average seasonality) and the interannual variation of the date of occurrence (measure of dispersion of events) of $Q_{10}$ and $Q_{90}$ (e.g., Blöschl et al., 2017;Laaha and Blöschl, 2006;Vlach et al., 2020). The approach of circular statistics converts Julian dates into angular values corresponding to locations on the circumference of a circle and avoids problems with calculating the mean date when the dates of occurrence fall around the end or the beginning of a calendar year (Chen et al., 2013;Young et al., 2000;Hanus et al., 2021). The calendar date of occurrence is converted to an angular value as follows:

$$\theta_i = D_i \cdot \frac{2\pi}{m_i}, \qquad 0 \le \theta_i \le 2\pi \tag{1}$$

where $\theta_i$ is the angular date of occurrence in radians, $D_i$ varies between 1 and 365 (366 for leap years) and corresponds to the Julian date of occurrence of the flow event (e.g. $Q_{10}$ or $Q_{90}$) in the calendar year i, and $m_i$ is the number of days in that year.

The average date of occurrence $\overline{D}$ is calculated as:

$$\overline{D} = \begin{cases} \tan^{-1}\left(\dfrac{\overline{y}}{\overline{x}}\right) \cdot \dfrac{\overline{m}}{2\pi}, & \overline{x} > 0, \overline{y} \geq 0 \\[2ex] \tan^{-1}\left(\dfrac{\overline{y}}{\overline{x}}\right) \cdot \dfrac{\overline{m}}{2\pi} + \pi, & \overline{x} \leq 0 \\[2ex] \tan^{-1}\left(\dfrac{\overline{y}}{\overline{x}}\right) \cdot \dfrac{\overline{m}}{2\pi} + 2\pi, & \overline{x} > 0, \overline{y} < 0 \end{cases} \tag{2}$$

with

$$(\overline{x}, \overline{y}) = \left(\frac{1}{n}\sum_{i=1}^{n}\cos\theta_i, \frac{1}{n}\sum_{i=1}^{n}\sin\theta_i\right) \tag{3}$$

$$\overline{m} = \frac{1}{n}\sum_{i=1}^{n}m_i \tag{4}$$

where $\overline{x}$ and $\overline{y}$ represent the cosine and sine components of the average date, respectively, $\overline{m}$ is the average number of days per year, and $n$ is the total number of years.

The concentration of the dates of occurrence around the average date is given by the mean resultant $R$, as follows:

$$R = \sqrt{\overline{x}^2 + \overline{y}^2}, \qquad 0 \leq R \leq 1 \tag{5}$$

When $R$ approaches 1, the timing of the flow event ($Q_{10}$ or $Q_{90}$) is highly seasonal (the events occur on the same day of the year), but a small value of $R$ near 0 indicates a high interannual variability of the date of occurrence (events are evenly distributed over the year).

**4.6 Changes in high and low flows**

The evolutions of the dates of occurrence and the concentration of the date of occurrence of $Q_{10}$ and $Q_{90}$ are illustrated in Figure 14 and Figure S57. The median date of occurrence of $Q_{10}$ ($D_{Q10}$) varies between the Julian calendar days 254 and 261 (second dekad of September) on average across the three sub-basins (Black Volta, White Volta, Oti) over the historical period, and it is projected to drop by -2 days under RCP2.6 and RCP4.5 and increase by +2 days under RCP8.5 over the twenty-first century. However, higher changes are projected in the Black Volta ($\Delta D_{Q10}$ = -4 days under RCP2.6), White Volta ($\Delta D_{Q10}$ = -6 days under RCP2.6) and the Oti ($\Delta D_{Q10}$ = +5 days under RCP8.5) over 2051-2080. The concentration of the date of occurrence ($R$) of $D_{Q10}$ shows a high seasonality in the occurrence of high flows ($R_{Q10}$ = 0.94) across sub-basins, which does not change considerably over the twenty-first century (Figure S57).

In contrast to $D_{Q10}$, the median $D_{Q90}$ varies between 126 and 132 (first to second dekad of May) over the historical period and rises on average by +5 days over future periods and across sub-basins. However, notable rises in $D_{Q90}$ are observed in each sub-basins during 2071-2100 as follows: Black Volta ($\Delta D_{Q90}$ = +9 days under RCP8.5), White Volta ($\Delta D_{Q90}$ = +11 days under RCP8.5) and Oti ($\Delta D_{Q90}$ = +10 days under RCP2.6), which might be explained by the forward shift of the rainy season. The median $R_{Q90}$ is 0.74 on average across sub-basins and slightly drops in the future, denoting a higher variation in the seasonality of low flows.

[Figure]

Figure 14. Mean Julian dates of occurrence (*D*) of annual high flows (*Q*$_{10}$) and low flows (*Q*$_{90}$) over the historical (1991-2020) and future periods in the major sub-basins of the VRB (Black Volta, White Volta, Oti).

[Figure]

Figure S57. Concentration of the dates of occurrence (R) around the average date for annual high flows (Q$_{10}$) and low flows (Q$_{90}$) over the historical (1991-2020) and future periods at selected streamflow gauges.

-Fig 11, 12, S38, S49 (AET in particular) are mostly dominated by climate gradient and not showing vegetation dynamics. The authors should find a way to exclude the dominant effect of rainfall using a normalization procedure. A new procedure is proposed in this paper

Example:

https://www.preprints.org/manuscript/202111.0225/v1

However, there must be other methods approaches in the literature for deblurred AET pattern maps by removing climate gradient.

https://www.researchgate.net/publication/338208138_Image_Deblurring_Techniques_-A_Detail_Review

We agree with the reviewer that actual evaporation (Ea) patterns are dominated by rainfall patterns in the Volta basin. The study area is located in a semi-arid zone where the main driver of the water cycle is rainfall, with annual Ea accounting for 80% of rainfall. Therefore, Ea naturally follows the pattern of rainfall, which can also be observed for vegetation, i.e. there is more vegetation in the southern part of the basin where rainfall is high and less as we move towards the north (Figure 1). The normalization of Ea would not bring substantial information to the reader as we are interested in analyzing the main hydrological variables and not their derivatives (e.g. the evaporative index, calculated as the ratio of Ea to P). We understand the potential value of the normalization for model calibration, as in the paper suggested by the referee (https://www.preprints.org/manuscript/202111.0225/v1), but the goal of our manuscript is not focused on calibration.

-Line 196: "Uncertainties in the model inputs and outputs are assessed in terms of variability be"

Indicating V2 estimation (or even COV coefficient of variation) as uncertainty assessment is quite ambitious without a systematic uncertainty propagation like in GLUE by Keith Beven.

In short, this vague sentence should be revised as smth like "variability in the model inputs and outputs are assessed using V2 statistics".

The sentence is now modified into "Variability in the model inputs and outputs resulting from different climate models are assessed using the second order coefficient of variation ($V_2$)".

**Reply to Anonymous Referee #2**

Dembele et al. (2021) evaluate uncertainty in hydrologic variables based on twelve GCMs from CMIP5 dynamically downscaled by five RCMs over the Volta River Basin in Africa. This paper is written clearly, and it is an interesting study, particularly considering the expected population doubling between 2010 and 2050 and implications, which can be associated with changes in water redistribution. I have some minor comments which I kindly ask the authors to consider during manuscript revisions.

We thank the reviewer for this very positive overall appreciation of our work and the constructive review and valuable comments. Below are our responses to each of the comments.

Some less critical analyses could possibly be moved to Supplement to ease the reading and to keep the focus on the most important results. Mainly, if a figure is explained in one sentence, it requires to be moved to Supplement, e.g. Fig.12). Possibly, the Supplement itself is also very lengthy, and I suggest, remove less important figures.

We agree with the reviewer to remove Fig.12 from the manuscript. Moreover, the supplementary material content has now been condensed and the file has now 33 pages instead of 58 pages in the previous version.

On the other hand, I have missed some evaluation of hydrologic model performance. I understand that was done in earlier studies, but it might be helpful to include observation-based climatology, say from ERA5, into Figure 5.

The paper is already long as the reviewer highlighted in the previous comment. Therefore, we prefer to refer the reader to the full study on model calibration where model performance is detailed.

However, for a quick look at the model performance, the reader is now referred to Figure 8 of Dembele et al. 2020b (see L175).

For consistency, adding ERA5 climatology to Figure 5 would require also adding the climatology of the other reference datasets used for bias-correction (i.e. PGF v3, WFDEI, EWEMBI, MERRA-2, JRA-55 TAMSAT v3.0, CHIRPS v2.0, ARC v2.0, MSWEP v2.2, PERSIANN-CDR v1r1), which would result in a glutted figure. We think Figure 3 already provides a good comparison of the satellite and reanalysis datasets to the RCM-GCM datasets.

By using the historical period 1991-2020, the historical and RCP simulations get mixed. Should not be the historical period be considered only prior 2005?

The historical period can be chosen beyond 2005 as done in some previous studies (e.g. Almazroui et al. 2021; Hanus et al. 2021; Mengistu et al. 2021; Abubakari et al. 2019). As said by Hawkins and Sutton (2016): "A number of factors enter the decision about an appropriate observational reference period, for example, to be representative of the most recent conditions but long enough not to be overly influenced by random fluctuations, to be a period the public can relate to…". According to Liersch et al. (2020): "The Intergovernmental Panel on Climate Change (IPCC) will use the years 1995–2014 in its Sixth Assessment Report". The most current and widely used reference period for climate analyses by the World Meteorological Organization (WMO) is 1981–2010. We have chosen the period 1991-2020 to have a more recent context for understanding climate change. We have now explained the choice of our reference period in the manuscript by adding the following sentence at lines 179-180: "The baseline or historical period for climate change impact assessment is 1991-2020, which is chosen to have a more recent context for understanding climate change (Hawkins and Sutton, 2016)".

The title requires changes. Please replace mainly these two words that do not fit the current version of the paper: "dynamics" and "processes".

We agree to substitute the term "dynamics" to "changes" in the title and where appropriate in the manuscript. We think that "hydrological processes" is still a valid wording as described at this web link https://www.nwrfc.noaa.gov/info/water_cycle/hydrology.cgi.

Which PET method was used in the climate projections? This study (e.g. https://www.nature.com/articles/nature11575) suggests that different results can be obtained for different PET methods). Could you please clarify?

We agree that different PET methods would lead to different results. In fact, we have thoroughly discussed the limitations of the choice of PET methods at lines 497-511 but we missed to mention the method we have used (i.e. Hargreaves and Samani). The following sentence is now added to the manuscript at line 499: "Here, the Hargreaves and Samani (1985) method is used to calculate potential evaporation"

Bootstrapping should be considered for the analysis presented in Fig. 9, Fig. 13, to account for varying sample sizes between RCPs.

We understand the suggestion of bootstrapping as we do not have the same number of models for each RCP. The bootstrapping can help to randomly select a common number of RCMs to do the analysis and repeat it. However, we think it will not solve the problem, as the underlying distributions will still not be the same. One solution would be to restrict these analyses to the models for which we have the 3 RCPs available but it would restrict our analysis to only 5 RCM_GCM runs. Therefore, we prefer keeping the results as it with as many RCM_GCMs as possible. However, we have now

highlighted the issue of varying sample sizes between RCPs by adding the following sentence at lines 471-472: "As the number of models varies among RCPs, bootstrapping could be used to randomly select a common number of models but that would limit our analyses to five RCM-GCM combinations, and might not lead to substantial changes in the results".
* * *
Textual suggestions:

lines 16-17: Rephrase abstract, the first sentence, into something like: "This study conducts a comprehensive evaluation of the impacts of climate change on the West Africa Volta River basin's water resources, as the region is expected to be hardest hit by global warming."

We prefer the current formulation.

lines 22-23: Reformulate into something like: "The bias-corrected climate projections are then used as input to the mesoscale Hydrologic Model (mHM) for hydrological projections over the twenty-first century (1991-2100)."

Done.

Lines 31-32: rephrase into: "and amplifying the local population's vulnerability."

"and amplifying the vulnerability of the local population." is preferred.

line 37: "at a faster rate" => "faster"

Done.

lines 43-44: "Climate change and anthropogenic pressures increase water resources' stress (Sood et al., 2013)"

Modified to "Climate change and anthropogenic pressures increase stress on water resources".

line 45: "for" => "to"

Done.

line 60: rephrase into "usually focused"

Done.

line 67: reformulate to "the repercussions"

Done.

line 72 remove "provide knowledge to"

Done.

line 75: maybe "central" instead of "major"

"major" is more appropriate here.

Figure 1: it would be more helpful to split the grand legend block into figure panels, where individual classes are shown.

Not sure if we understood the suggestion. However, the current figure arrangement allows saving space and avoids not having many figures only for the description of the study area.

Line 94: "the assessment of" into "assessing"

Done.

Line 123-124: Possibly rephrase into "As the RCMs downscale not all GCMs,…"

Ok. Would be "As the RCMs do not downscale all the GCMs…".

Line 188: "a steady-state"

Done.

Figure 3 caption: Write clearly this is the historical period (keep consistency with the other figures)

Done.

Figure 5 caption: should be: "… for the historical and future periods (under RCP8.5)."

Done.

Figure 6-7 caption: synchronize legend (e.g., *_2050) with figure caption (2021-2050).

Done.

**References**

Abubakari, S., Dong, X., Su, B., Hu, X., Liu, J., Li, Y., … & Xu, S. (2019). Modelling streamflow response to climate change in data-scarce White Volta River basin of West Africa using a semi-distributed hydrologic model. Journal of Water and Climate Change, 10(4), 907-930. https://doi.org/10.2166/wcc.2018.193

Almazroui, M., Ashfaq, M., Islam, M. N., Rashid, I. U., Kamil, S., Abid, M. A., … & Sylla, M. B. (2021). Assessment of CMIP6 Performance and Projected Temperature and Precipitation Changes Over South America. Earth Systems and Environment, 1-29. https://doi.org/10.1007/s41748-021-00233-6

Hanus, S., Hrachowitz, M., Zekollari, H., Schoups, G., Vizcaino, M., and Kaitna, R. (2021). Future changes in annual, seasonal and monthly runoff signatures in contrasting Alpine catchments in Austria, Hydrology and Earth System Sciences, 25, 3429-3453, 645 https://doi.org/10.5194/hess-25-3429-2021

Hawkins, E., & Sutton, R. (2016). Connecting climate model projections of global temperature change with the real world. Bulletin of the American Meteorological Society, 97(6), 963-980. https://doi.org/10.1175/BAMS-D-14-00154.1

Liersch, S., Drews, M., Pilz, T., Salack, S., Sietz, D., Aich, V., … & Hattermann, F. F. (2020). One simulation, different conclusions—the baseline period makes the difference!. Environmental Research Letters, 15(10), 104014. https://iopscience.iop.org/article/10.1088/1748-9326/aba3d7

Mengistu, D., Bewket, W., Dosio, A., & Panitz, H. J. (2021). Climate change impacts on water resources in the Upper Blue Nile (Abay) River Basin, Ethiopia. Journal of Hydrology, 592, 125614. https://doi.org/10.1016/j.jhydrol.2020.125614

---

## Author Response (AR2)

**Reply to the Editor and the Referees of the manuscript**
**"Contrasting changes in hydrological processes of the Volta River basin under global warming"**
**(Manuscript hess-2021-525)**
**by M. Dembélé et al.**

**Reply to the Editor**

Based on your responses and the carefully revised manuscript, I am pleased to accept your paper for publication pending a few minor corrections.

We thank the editor for handling our manuscript and for accepting to publish it after minor corrections. Below are our answers to the minor corrections.

Figure 1: the symbology of the DEM can be updated to make it more intuitive. The colour scheme should be inverted to make the lower ground more blueish and higher elevation more brownish.

I also note Figure 1 is exactly the same as what you used in your previous paper e.g. 2020b. Please check if you have the copyright to reuse the published material in this manuscript.

Our previous paper is published in WRR (https://doi.org/10.1029/2019WR026085), an AGU journal. According to the AGU policy, we have the right to reuse our published figure (see https://www.agu.org/Publish-with-AGU/Publish/Author-Resources/Policies/Permission-policy). However, we have now added the mention "Adapted from Dembélé et al., 2020b" and made some modifications to the figure to invert the colour scheme of the DEM as requested.

[Figure]

Line 271: shifted towards a lower evaporative index and they large model dependent variability in aridity ranges

Change to '… and larger model dependent variability in the aridity index ranges'

Corrected to: "…and they have larger model-dependent variability in aridity ranges"

Figure 5: please confirm if the unit of each variable shown in the figure is a monthly value. If so, why is it St (mm) and Su (mm/mm)? If they are all monthly values, you can remove '/month' from the unit of all the variables. Please also check the other similar figures in the SI and make appropriate correction.

Yes, the unit of each variable in Figure 5 is a monthly value. Here, and in the entire manuscript we make a difference between the units of variables that represent fluxes or flows (i.e. dimension is [L/T]) and units of state variables (i.e. dimension is [L]). See for instance, Table 1 of Bouaziz et al. 2021 (https://doi.org/10.5194/hess-25-1069-2021). Rainfall, evaporation, runoff and recharge are fluxes (i.e. mm/month), while temperature, soil moisture (Su) and terrestrial water storage (St) are state variables (i.e. °C or mm). In addition, Su is given with a volumetric unit (i.e. mm/mm) corresponding to the amount of water per depth of soil, as simulated by the mHM model (see e.g. Figure 3 at https://doi.org/10.5194/hess-2021-402), and also Figure 1 at https://crops.extension.iastate.edu/blog/mark-licht-mike-castellano-sotirios-archontoulis/facts-soil-moisture-benchmarking-tool).

Line 339-340: The largest uncertainty in annual climate projections over the historical period is observed for rainfall with a V2 of 3.2% for the inter-model variability.

Please refer to the figure number as it is not from Figure S32 and check if 3.2% is correct.

Actually, V2 of 3.2% is the variability among the median of rainfall projections over the historical period under different RCPs (Figure S32). This sentence is removed to avoid confusions.

Line 523: Here, the Hargreaves and Samani (1985) method is used to calculate potential evaporation.

This should be placed in the method section.

Done. Now placed in section 3.4.1.